# 🤖 MLLM-ISU: The First-Ever Comprehensive Benchmark for Multimodal Large Language Models based Intrusion Scene Understanding

**Fujun Han**[1], **Peng Ye**[2,3]*

[1] School of Data Science, The Chinese University of Hong Kong, Shenzhen
[2] Shanghai AI Laboratory    [3] MMLab, The Chinese University of Hong Kong
hanfujun@cuhk.edu.cn, yepeng@pjlab.org.cn

## Abstract

Vision-based intrusion detection has multiple applications in practical scenarios, *e.g.*, autonomous driving, intelligent monitoring, and security. Previous works mainly focus on improving the intrusion detection performance, without a comprehensive and in-depth understanding of the intrusion scene. To fill this gap, we explore a novel task called Multimodal Large Language Models based Intrusion Scene Understanding (MLLM-ISU) and report a comprehensive benchmark for the task. Specifically, we first design an effective and automatic visual question-answer generation strategy, constructing a new MLLM-ISU dataset, with 3000 VQA evaluation Pairs, 8925 training Pairs, and six relevant subtasks. Then, we perform a comprehensive assessment on various state-of-the-art proprietary and open-source MLLMs, *e.g.*, DeepSeek-VL2, GPT-4o, Qwen2.5-VL, *etc*, and find that current MLLMs have weak abilities for this task. Further, in order to improve the intrusion understanding capabilities of current MLLMs, we propose a Post-Training Framework with three sequential training stages, *i.e.*, Intrusion-aware Visual Instruction Pre-training, Intrusion Chain of Thought tuning, and Intrusion-centric VQA tuning, and sufficient experiments and comparisons are conducted to verify the effectiveness of the proposed three-stage training framework. Available datasets and codes: https://github.com/1012537710/MLLM-ISU.

## 1 Introduction

Vision-based intrusion detection has widespread applications in multiple domains, *e.g.*, autonomous driving, intelligent monitoring, and security. Vision-based intrusion detection refers to judging whether a possible object exists in the restricted area-of-interest (AoI), and existing works can be divided into two main categories via whether the camera is moving or not, *i.e.*, static-view intrusion detection [41, 25, 27] and dynamic-view intrusion detection [28, 11, 12, 10]. However, these works can only perform simple intrusion detection tasks and cannot achieve comprehensive and in-depth intrusion scene understanding, *i.e,* intrusion summary analysis, intrusion detection descriptions, as shown in Tab. 1. And previous intrusion models cannot fulfill detailed descriptions like humans. Besides, when the environment or scene changes, previous works struggle to retrain the network so that the model can adapt to the new scene's needs.

With the continuous development of Multimodal Large Language Models (MLLMs) in recent years, *i.e.*, Deepseek [9, 21, 8], ChatGPT [1], LLaVa [22], Qwen [4], and Gemini 1.5 [29], advanced scene understanding seems to be possible. Thus, we cannot help but ask the basic yet important

---

*Corresponding Author

39th Conference on Neural Information Processing Systems (NeurIPS 2025) Track on Datasets and Benchmarks.

question: *How about the capability of current MLLMs in Intrusion Scene Understanding?* Although MLLMs have showcased outstanding performance in multiple fields such as *e.g.*, Fine-Grained Video Motion Understanding [32], Industrial Anomaly Detection [20], the capability of intrusion scene understanding still remains **blank** and **unexplored**. To remedy the limitations mentioned above and answer this question, we define and investigate a new and practical task, Multimodal Large Language Models based Intrusion Scene Understanding (**MLLM-ISU**) for the first time.

To accomplish the aforementioned MLLM-ISU task, the first challenge is the lack of relevant datasets to assess the ability of existing MLLMs to understand the intrusion scenes. Previous related datasets, *e.g.*, Cityintrusion [28], Cityintrusion-Multicategory [11], and Multi-Domain Multi-Category [12], mainly focus on image understanding of structured outputs, *e.g.*, object detection and semantic segmentation, and lack the capabilities of visual reasoning guided by linguistic instructions. In this paper, we design and introduce the first MLLM-ISU dataset, with 3000 VQA Pairs for evaluation, 8925 Pairs for supervised fine-tuning training, and six relevant subtasks, *i.e.*, two low-level understanding subtasks: Intrusion Behavior Judgment, Person Intrusion Classification, three high-level understanding subtasks: Intrusion Summary Analysis, Intrusion Object Localization, Intrusion Category Identification, and an open-level understanding subtask: Intrusion Scene Descriptions. We show that our MLLM-ISU dataset can effectively meet the requirements of the proposed task.

The second challenge is that there is still a lack of effective training strategies to improve the capability of current MLLMs in the intrusion scene understanding task. To address this, we propose a new post-training framework with three sequential training stages: an Intrusion-aware Visual Instruction Pre-training to get the capability of basic coarse-grained scene understanding; an Intrusion Chain of Thought Tuning strategy to give the model the ability to reason and explain; an intrusion-centric VQA tuning method to enhance fine-grained and structured understanding skills. Comprehensive experiments show that our framework can effectively improve the intrusion scene understanding ability of current MLLMs.

Table 1: The comparison between previous promising intrusion detection and our MLLM-ISU task. 🔴 and 🟢 denote the low and high scene understanding capabilities of different intrusion models.

| Intrusion tasks | Task Type | Structure | Modal | Scene Understanding (↑) |
|---|---|---|---|---|
| PIDNet [28] | Detectable Single Task | Close | Single-Modal | 🔴 |
| MF-ID [11] | Detectable Single Task | Close | Single-Modal | 🔴 |
| MM-ID [12] | Detectable Single Task | Close | Single-Modal | 🔴 |
| Ada-iD [10] | Detectable Single Task | Close | Single-Modal | 🔴 |
| MLLM-ISU | Comprehension Multi Task | Open | Multi-Modal | 🟢 |

Our contributions can be summarized as: (1) **Novel task and benchmark.** To the best of our knowledge, the MLLM based Intrusion Scene Understanding (MLLM-ISU) task is proposed for the first time. A comprehensive benchmark, including relative datasets and baselines, is reported for the specific task. All datasets, codes, and baselines will be publicly available. (2) **Effect strategies and framework**. We introduce an effective automatic visual question-answer generation pipeline, overcoming the limitation that existing datasets are not directly applicable to the MLLM-ISU task. Besides, an effective post-training framework with sequential training strategies is designed to improve the performance of the MLLM-ISU task. (3) **Sufficient experiments**. We conduct comprehensive experiments to evaluate the intrusion understanding capability of current MLLMs and demonstrate the effectiveness of the proposed framework. Our framework can effectively improve the weak comprehension of current MLLMs, even surpassing the proprietary MLLMs.

## 2  Related Work

**Traditional Vision-based Intrusion Detection.** Traditional Vision-based Intrusion Detection tasks mainly focus on two main directions: static-view and dynamic-view intrusion detection. For static-view intrusion detection tasks, some promising methods, *e.g.*, Histogram of Oriented Gradients (HOG) [41], Conditional Random Field (CRF) [25], and Adaptive Background Subtraction (ABS) [27], are proposed to solve the problem in a static environment. Besides, to compensate for the lack of generalizability and category diversity of previous static-view works in adverse or challenging environments, some encouraging dynamic-view works and methods, *e.g.*, MF-ID [11], MMID (Unsupervised Domain Adaptation) [12], Ada-iD (Active Domain Adaptation) [10], are proposed to improve the performance of different intrusion detection frameworks in dynamic-view. However, although these works effectively address the problem of dynamic-view intrusion detection, they can only accomplish the single task of detection and not the multi-task of intrusion scenario understanding, as shown in Tab. 1. Thus, in this paper, we propose a new and vital task to solve the problem, *i.e.*, Multimodal Large Language Models based Intrusion Scene Understanding.

**Multimodal Large Language Models Benchmarks.** Multimodal Large Language Models achieve promising performance in multiple understanding, and rich benchmarks are proposed. On the one hand, some datasets are designed to evaluate the comprehensive performance of LLMs or MLLMs, *e.g.* Math/Science (GSM8k [6], MATH [16], MathQA [2]), Reasoning (CLRS [33], HiPho [39]), Image Understanding (MMMU [40], MathVista [23], ChartQA [24] ), Coding (DeepSeek-Coder [9]), *etc*. Based on these datasets, the comprehensive ability of understanding is also reported. On the other hand, for specific tasks, richer baselines are established and given, *e.g.*, Fine-Grained Video Motion Understanding (FAVOR-Bench) [32], Olympiad-level bilingual multimodal scientific (OlympiadBench) [14], and End-to-End Autonomous Driving (OpenEMMA) [36], Enhanced MultiModal ReAsoning (EMMA) [13], Industrial Anomaly Detection [20]. However, although these promising benchmarks are proposed, the capability of intrusion scene understanding still remains *blank* and *unexplored*. Therefore, to remedy the limitation, in this paper, we propose a comprehensive benchmark for the MLLM-ISU task, including relevant datasets, framework, and baselines.

## 3 MLLM-ISU Dataset

### 3.1 Data Collections

Vision-based intrusion detection (V-ID) is a joint task with detection and segmentation. Therefore, the V-ID task usually needs both detection and segmentation annotations. However, acquiring these labels for the same image is very time-consuming and difficult. Fortunately, with the development of computer vision, some promising vision-based intrusion detection works are proposed, *e.g.*, PIDNet [28], MF-ID [11], MMID [12], and Ada-iD [10]. By investigating, we find that the designed datasets for the specific intrusion detection works are based on the Cityscape dataset [7], *e.g.*, Cityintrusion [28], Cityintrusion-Multicategory [11], and Multi-Domain Multi-Category [12]. The main reason is that the Cityscape dataset provides rich detection and segmentation annotation on the same original image, making it the dataset of choice for intrusion detection tasks. Therefore, in this paper, we also use the Cityscape dataset to build our MLLM-ISU datasets and further test the capability of current MLLMs. Our MLLM-ISU datasets include 3000 VQA Pairs for evaluation, with six relative subtasks, as shown in Section 3.2. Besides, we also design 8925 training Pairs to help improve intrusion understanding performance for MLLMs, and the details are shown in Section 4. The detailed data statistics and word cloud of all questions are shown in Fig. 1 and Fig. 2, respectively. Note that to verify the universality of the proposed pipeline of VQA-Data Generation and enhance the diversity of intrusion scene types in real-world environments, we create a new benchmark dataset based on the BDD-100K [38] for the MLLM-ISU task. Please refer to the section Discussion.

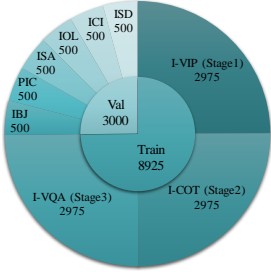
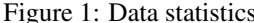

Figure 1: Data statistics

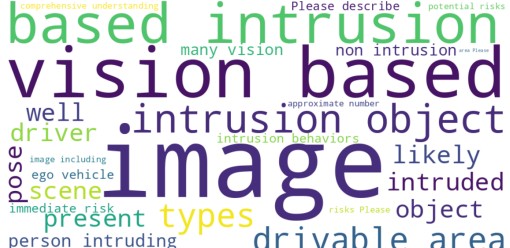

Figure 2: The word Cloud of all questions

### 3.2 Question Definition

To evaluate the comprehensive intrusion scene understanding capability of current MLLMs, we define six different intrusion subtasks from multiple views: *low-level understanding tasks* (L-U-T), *high-level understanding tasks* (H-U-T), and *open-level understanding tasks* (O-U-T), respectively. The L-U-T mainly focuses on simple intrusion understanding and contains two subtasks: Intrusion Behavior Judgment and Person Intrusion Classification. Besides, H-U-T focuses on high-level understanding and contains three subtasks: Intrusion Summary Analysis, Intrusion Object Localization, and Intrusion Category Identification. O-U-T is an open understanding task and contains the Intrusion Scene Descriptions subtask. Each subtask is designed to test different capabilities of MLLMs, as follows.

· **Subtask1: Intrusion Behavior Judgment (IBJ)**. The subtask of IBJ is used to test whether the MLLMs can correctly make intrusion judgments. It is a binary classification and can measure the model's ability to determine intrusion events.

· **Subtask2: Person Intrusion Classification (PIC)**.The subtask of PIC is designed to test the model's intrusion judgments for individual categories. It is used to measure the model's ability to categorize intrusion events at a *fine-grained* level.

· **Subtask3: Intrusion Summary Analysis (ISA)**. ISA is used to test the ability to summarize and analyze in the intrusion scene understanding task, *i.e.*, how many intrusion and non-intrusion behaviors are in the image.

· **Subtask4: Intrusion Object Localization (IOL)**. IOL is designed to measure the localization capabilities of MLLMs. This subtask is used to determine the object location for different intrusion behaviors and to identify the most dangerous intrusion objects.

· **Subtask5: Intrusion Category Identification (ICI)**. This subtask is used to test the ability to identify the intrusion category. Ask the MLLMs which types have intruded into the divine area. about the type of intrusion detection behavior.

· **Subtask6: Intrusion Scene Descriptions (ISD)**. This is an open-level question that aims to assess the ability to describe intrusion detection scenes.

### 3.3 VQA-Data Generation

**Automatic Question-Answer Generation.** To obtain the final **VQA** pairs, we design a novel and efficient data generation strategy. Compared to some automatic Question-Answer generation methods by using MLLMs (GPT-4V or GPT-4o), *e.g.*, MMAD [20], FAVOR-Bench [32], Biology Instructions [15], our proposed strategy presents more efficiency and accuracy, as shown in Fig. 3. Specifically, our data generation strategy mainly contains four main steps: (i) We first utilize the detection labels (from the XML file) and segmentation labels in the original cityscape dataset [7] to compute overlapping pixel points. (ii) Based on the overlapping pixel point results, we obtain intrusion ('Y') and non-intrusion ('N') labels. Following the previous promising work [28, 11, 10], we set the threshold of overlapping pixel points to **20**. (iii) Then, we write the obtained intrusion ('Y') and non-intrusion ('N') labels to the original XML file. The new XML file contains three rich labels for the intrusion scene understanding task, *i.e.*, the intrusion/non-intrusion categories, bbox coordinates, and intrusion/non-intrusion labels. (iv) Finally, the original image and the newly obtained XML file are utilized to generate VQA pairs for multiple subtasks. Note that in order to test the real capabilities of the MLLMs better, we set the *Random* method to avoid biased/tendentious answers for some subtasks and to distribute the answers for all options as much as possible.

**Manual Verification.** After getting a preliminary VQA pair, a team of multiple students is formed to manually check to make sure all questions and answers are aligned and correct. Besides, for the

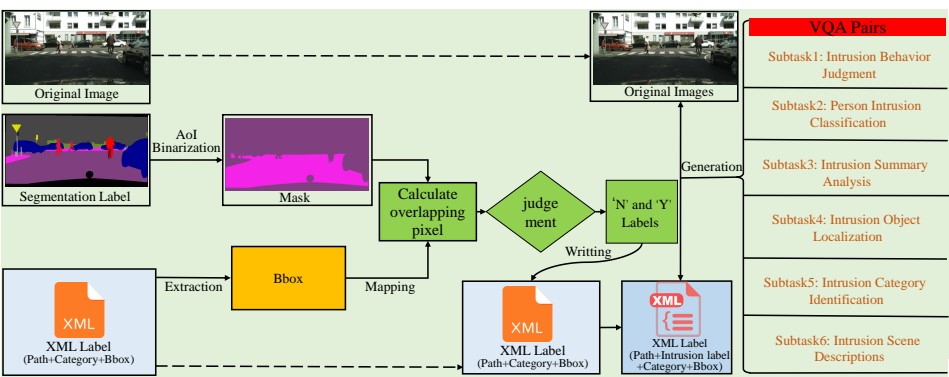

Figure 3: The pipeline of efficient data generation strategy. Step 1: Use the segmentation label to get the AoI mask and read the Bbox of the XML document. Step 2: Calculate the overlapping pixels and give the Intrusion/Non-intrusion labels. Step 3: Write the intrusion labels for the original XML to get the new XML. Step 4: Combine the original image and new XML to generate final VQA Pairs.

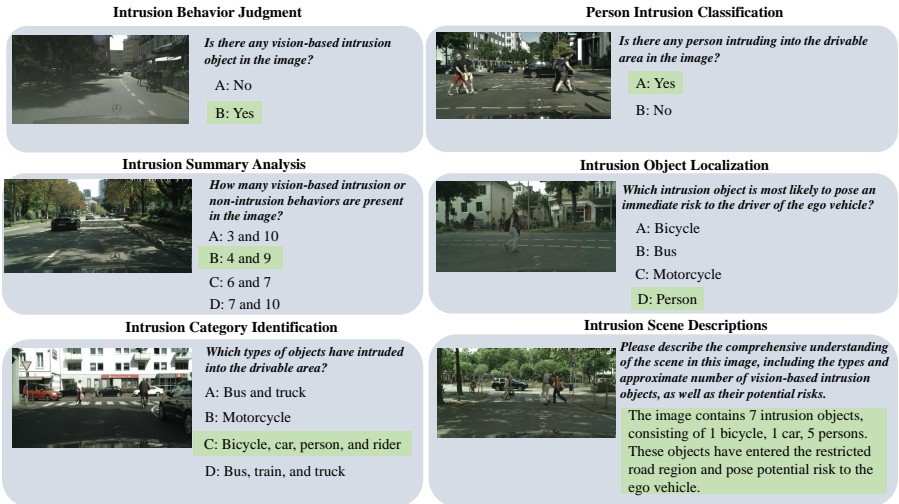

Figure 4: The detailed cases of the proposed VQA evaluation pairs in the MLLM-ISU dataset.

VQA task, we add some content to the prompt to get the correct structured answers, *e.g.*, <Please answer with only A or B>. Our final Question-Answer pairs for intrusion scene understanding are shown in Fig. 4. More VQA Pairs cases can be found in **Appendix A**.

# 4 Three-stages Post-Training Framework

To improve the capability of MLLMs in the intrusion scene understanding task, we propose a three-stage post-training framework with three different Supervised Fine-tuning strategies, *i.e.*, Perception (Intrusion-aware Visual Instruction Pre-training)→Reasoning (Intrusion Chain of Thought Tuning)→Understanding (Intrusion-centric VQA Tuning). The detailed workflow is shown in Fig. 5.

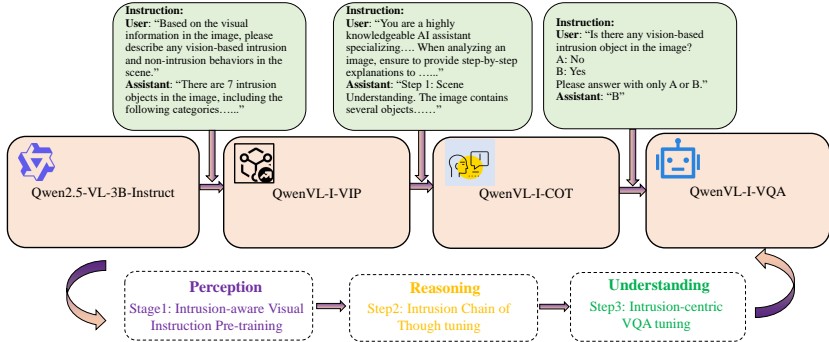

Figure 5: The workflow of the three-stage post-training. We take the Qwen2.5-VL-3B-Instruct as an example. For MLLMs, we first propose an Intrusion-aware Visual Instruction Pre-training strategy to obtain the initial coarse-grained perception capabilities. Then, an Intrusion Chain of Thought Tuning method is designed to enhance the ability to reason. Finally, we introduce an Intrusion-centric VQA Tuning to further enhance the capability of understanding fine-grained structures.

## Stage 1: Intrusion-aware Visual Instruction Pre-training (I-VIP)

To give initial perception capabilities to MLLMs in the vision-based intrusion scene understanding task, we propose an Intrusion-aware Visual Instruction Pre-training strategy. Specifically, we use the system prompt: "*Based on the visual information in the image, please describe any vision-based intrusion and non-intrusion behaviors in the scene*". This prompt will help the MLLMs to understand the basic intrusion and non-intrusion behaviors and have a coarse-grained scene understanding. Besides, to efficiently perform supervised fine-tuning of the original MLLMs, we adopt the Lora [17] Supervised Fine-Tuning (SFT) method. The main reason is that Lora Supervised Fine-Tuning (SFT)

received great attention due to its efficiency and effectiveness strategy in improving the performance of MLLMs [15, 31]. Our detailed prompt template is shown in **Appendix B**.

**Stage 2: Intrusion Chain of Thought Tuning (I-COT)**

Based on Stage 1, the MLLMs obtain effective coarse-grained (initial perception) capabilities. However, the models still have weaknesses in structured reasoning aspects. Therefore, in order to make the model get the ability to reason and explain, *e.g.*, "what is intrusion and why it constitutes an intrusion". We design a novel Intrusion Chain of Thought Tuning approach to improve the performance of the scene understanding. We use the system prompt <*You are a highly knowledgeable AI assistant specializing in vision-based intrusion detection scene understanding task. When analyzing an image, ensure to provide step-by-step explanations to make your*

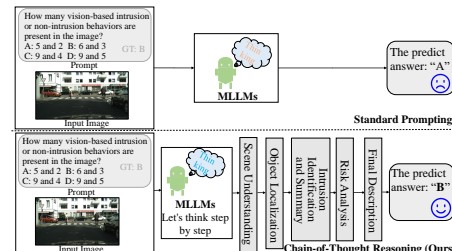

Figure 6: The proposed pipeline of I-COT.

*responses correct and easy to understand*> and five reasoning steps to conduct the tuning. After I-COT, the reasoning process is shown in Fig. 6. We compare the original standard prompting strategy.

**Stage 3: Intrusion-centric VQA Tuning (I-VQA)**

In the aforementioned two stages, to further improve MLLMs' ability to understand and answer structured questions about different intrusion understanding tasks, we design six different subtasks and build a training VQA tuning dataset, containing 2975 VQA pairs. The VQA tuning dataset can go deeper to enhance the structured understanding of the model. Through VQA Supervised Fine-Tuning (SFT), MLLMs develop the skill of fine-grained understanding and can have a structured understanding of the different subtasks.

# 5 Experiment and Results

## 5.1 Experiment settings

**Benchmark Evaluation Models.** To evaluate and report the comprehensive performance of our proposed MLLM-ISU task, we conduct sufficient experiments and comparisons on some dominant MLLMs, *i.e.*, Proprietary and Open-source MLLMs. For proprietary MLLMs, because the original model is not available, we adopt accessible APIs to test, *e.g.*, Gemini-1.5-pro-latest [29], GPT-4o [19], and Claude-3.7-sonnet [3]. For Open-source MLLMs, we choose the Qwen-series models (*i.e.*, Qwen2-VL-Instruct [34], Qwen2.5-VL-Instruct-3B [4], Qwen2.5-VL-Instruct-7B [4]), LLava-series models (*i.e.*, LLava-1.5-7b-hf [22], LLava-1.5-13b-hf [22]), InternVL-series (*i.e.*, InternVL2.5-1B [5], InternVL2.5-2B [5], InternVL2.5-8B [5]), and the latest InterVL3-2B [5], InterVL3-8B [5], Gemma-series (Gemma3-4B-it [30] and Gemma3-12B-it [30]), DeepSeek-VL2-tiny [35], DeepSeek-VL2-small [35], Kimi-VL-A3B-Instruct [31], Kimi-VL-A3B-Thinking [31], MiniCPM-V2.6 [37]. Unless specified, all experiments will be evaluated in a zero-shot manner.

**Evaluation Metrics.** In proposed MLLM-ISU task, *low-level understanding tasks* (L-U-T) and *high-level understanding tasks* (H-U-T) are designed to as multiple-choice. Therefore, the evaluation metrics of these subtasks are set to **Accuracy**, *i.e.*, the proportion of options made correctly. For *open-level understanding tasks* (O-U-T), we use the **BLEU-4** (BiLingual Evaluation Understudy) [26] metrics to evaluate the performance. More metrics and results can be found in **Appendix C**.

## 5.2 Main Results and Findings

We conduct a comprehensive experiment on the proposed task using various MLLMs. The detailed evaluation results are shown in Tab. 2. Three main findings can be observed.

**Findings 1: The performance and comparison of different MLLMs**. 1) In our proposed task, the best overall performance is Qwen2.5-VL-3B-Instruct, and its performance can reach 50.79%. Note that the performance is even 8.74% higher than the Qwen2.5-VL-7B-Instruct. The main reason is that the Qwen2.5-VL-7B-Instruct model exhibits a weak ability in the intrusion judgment task, *i.e.*, subtask1: IBJ. The second-best performance model is Deepseek-VL2-small, the performance can reach 50.15%. 2) Besides, we can also find that different MLLMs exhibit different strengths

Table 2: Comprehensive performance evaluation of the 20 dominant MLLMs on the proposed task. The **bold** and underlined results denote the best and second-best performance.

| Model | Source | Release | L-U-T (Easy) | | H-U-T (Difficult) | | | O-U-T (Open) | Average |
|---|---|---|---|---|---|---|---|---|---|
| | | | IBJ | PIC | ISA | IOL | ICI | ISD | |
| Human | - | - | 98.00 | 72.00 | 44.00 | 77.00 | 98.00 | 39.36 | 71.39 |
| *Proprietary MLLMs* | | | | | | | | | |
| GPT-4o [19] | OpenAI | 2024-08 | 41.28 | 68.27 | 29.38 | 53.23 | 83.80 | 6.74 | 47.12 |
| Gemini-1.5-pro-latest [29] | Google | 2024-04 | 7.40 | 65.40 | **37.00** | 68.40 | 68.80 | 6.04 | 42.17 |
| Claude-3.7-sonnet [3] | Anthropic | 2025-02 | 56.14 | 56.40 | 20.28 | 53.71 | 52.00 | 9.09 | 41.27 |
| *Open-source MLLMs* | | | | | | | | | |
| LLaVa1.5-7B-hf [22] | UW-M&Micro | 2023-10 | **94.00** | 41.80 | 22.60 | 26.60 | 33.80 | 12.12 | 38.49 |
| LLaVa1.5-13B-hf [22] | UW-M&Micro | 2023-10 | 71.00 | 62.20 | 23.60 | 30.00 | 51.80 | 12.33 | 41.82 |
| Qwen2-VL-7B-Instruct [34] | Alibaba | 2024-06 | 56.20 | 65.60 | 33.80 | 52.20 | 57.80 | 5.93 | 45.26 |
| MiniCPM-V2.6 [37] | OpenBMB | 2024-08 | 7.00 | 64.80 | 32.60 | 38.00 | 81.00 | 11.65 | 39.18 |
| InternVL2.5-1B [5] | OpenGVLab | 2024-12 | 22.80 | 59.60 | 30.60 | 27.60 | 48.00 | 12.87 | 33.58 |
| InternVL2.5-2B [5] | OpenGVLab | 2024-12 | 7.60 | 60.60 | 30.40 | 44.80 | 61.60 | 12.53 | 36.26 |
| InternVL2.5-8B [5] | OpenGVLab | 2024-12 | 28.40 | 64.60 | 28.80 | 56.80 | 80.00 | 15.27 | 45.65 |
| DeepSeek-VL2-tiny [35] | DeepSeek | 2024-12 | 64.60 | 65.40 | 32.40 | 41.80 | 73.80 | **18.79** | 49.47 |
| DeepSeek-VL2-small [35] | DeepSeek | 2024-12 | 40.60 | 64.00 | 32.80 | 74.40 | 81.00 | 8.12 | 50.15 |
| Qwen2.5-VL-3B-Instruct [4] | Alibaba | 2025-01 | 43.00 | 61.20 | 30.20 | **79.80** | 84.20 | 6.36 | **50.79** |
| Qwen2.5-VL-7B-Instruct [4] | Alibaba | 2025-01 | 16.00 | 61.40 | 24.00 | 64.80 | 81.40 | 4.68 | 42.05 |
| Gemma3-4B-it [30] | Google | 2025-03 | 52.60 | 64.20 | 25.20 | 50.40 | 54.60 | 5.93 | 42.16 |
| Gemma3-12B-it [30] | Google | 2025-03 | 58.20 | 54.80 | 26.00 | 65.40 | 81.00 | 5.73 | 48.52 |
| Kimi-VL-A3B-Instruct [31] | Moonshot AI | 2025-04 | 8.60 | **74.20** | 23.40 | 47.80 | **87.20** | 11.65 | 42.14 |
| Kimi-VL-A3B-Thinking [31] | Moonshot AI | 2025-04 | 45.40 | 41.00 | 21.40 | 25.20 | 23.40 | 5.22 | 26.97 |
| InternVL3-2B [5] | OpenGVLab | 2025-04 | 24.40 | 63.60 | 35.20 | 41.40 | 85.80 | 6.49 | 42.82 |
| InternVL3-8B [5] | OpenGVLab | 2025-04 | 12.40 | 70.60 | 34.20 | 36.40 | 63.00 | 6.58 | 37.20 |

and weaknesses in multiple subtasks, *e.g.*, two low-level understanding tasks IBJ and PIC, the best performance reaches 94.00% (LLaVa1.5-7B-hf) and 74.20% (Kimi-VL-A3B-Intruct). The performance is relatively high. The main reason is that the two subtasks are binary classification tasks and are relatively simple. Therefore, the models can easily make accurate judgments. 3) In proprietary MLLMs, the best performance is 47.12% of GPT-4o, 3.67% below Qwen2.5-VL-3B-Instruct.

**Findings 2: Current MLLMs are not capable of intrusion scene understanding**. Although most of the MLLMs show promising performance on easy binary classification tasks, their performance does not seem good on some difficult and open tasks, *e.g.*. Intrusion Summary Analysis (ISA) subtask and Intrusion Scene Descriptions (ISD) subtask. The results denote that they are less capable of summarizing and analyzing visual intrusion detection, which is a direction worth improving subsequently. Note that in the open task Intrusion Scene Descriptions that we designed, all the models exhibit poor performance. This is mainly due to the models' insufficient ability to understand intrusion events and make correct judgments.

**Findings 3: The best MLLMs still have a wide gap with human performance.** To better evaluate the capability of current MLLMs in proposed tasks, we give comparisons to human performance and find that even the best models [4] still have a wide gap with human, *i.e.*, 50.79% vs 71.39%. This suggests a need to better enhance the intrusion scene understanding capability of current MLLMs.

### 5.3 Validation Experiments and Analyses

**Training settings**. To verify the effectiveness of the proposed improvement strategies, we conduct comprehensive experiments and analyses. Two promising and typical MLLMs are selected to train and evaluate, *i.e.*, Qwen2.5-VL-3B-Instruct and Qwen2.5-VL-7B-Instruct. Besides, we report the performance of training by the proposed six intrusion scene understanding subtasks, with two metrics (*i.e.*, Accuracy and BLEU-4). More training details are shown in **Appendix D**.

**Can the proposed three-stage post-training strategy be effective in improving the understanding capacity of MLLMs?** We first test the performance of the proposed three-stage post-training strategy, as shown in Tab. 3. We can find that when three different supervised Fine-tuning training strategies are added, the scene understanding capabilities of different MLLMs will be improved effectively, *e.g.*, Qwen2.5-VL-3B-Instruct: 50.79→52.04→55.50→72.15, Qwen2.5-VL-7B-Instruct:

Table 3: The performance of the proposed three post-training stages on different MLLMs. I-VIP, I-COT, and I-VQA denote the proposed three different strategies in the training stages, respectively.

| Model+Method | IBJ | PIC | ISA | IOL | ICI | ISD | Average |
|---|---|---|---|---|---|---|---|
| *3B Open-source MLLMs, Epoch=2* | | | | | | | |
| Qwen2.5-VL-3B-Instruct | 43.00 | 61.20 | 30.20 | 79.80 | 84.20 | 6.36 | 50.79 |
| Qwen2.5-VL-3B-Instruct+I-VIP | 49.00 | 61.60 | 28.00 | 81.00 | 86.60 | 6.03 | 52.04 |
| Qwen2.5-VL-3B-Instruct+I-VIP+I-COT | 52.20 | 58.60 | 25.00 | 83.60 | 89.60 | 24.00 | 55.50 |
| Qwen2.5-VL-3B-Instruct+I-VIP+I-COT+I-VQA | 94.60 | 67.20 | 36.80 | 88.80 | 97.20 | 48.31 | **72.15** |
| *7B Open-source MLLMs, Epoch=5* | | | | | | | |
| Qwen2.5-VL-7B-Instruct | 16.00 | 61.40 | 24.00 | 64.80 | 81.40 | 4.68 | 42.05 |
| Qwen2.5-VL-7B-Instruct+I-VIP | 85.60 | 63.40 | 23.20 | 81.40 | 91.60 | 30.51 | 62.62 |
| Qwen2.5-VL-7B-Instruct+I-VIP+I-COT | 95.40 | 66.00 | 24.60 | 82.00 | 95.20 | 54.37 | 69.60 |
| Qwen2.5-VL-7B-Instruct+I-VIP+I-COT+I-VQA | 95.80 | 78.40 | 53.60 | 92.60 | 99.40 | 50.44 | **78.37** |

42.05→62.62→69.60→78.37, which benefits from the advantage of supervised fine-tuning. Besides, in multiple subtasks, the performance of MLLMs is improved, which verifies the effectiveness of the proposed three-stage post-training strategy. We provide a new paradigm for training and improving performance for different tasks. More metrics and results can be found in **Appendix E**.

**What role does each stage play for performance in the MLLM-ISU task?** To further explore the role of every stage, we give the training loss, as illustrated in Fig. 7. We can find that in the early training steps, the loss significantly reduces, which indicates that the model rapidly learn the basic knowledge, *i.e.*, coarse-grained understanding capability. Then, by stage 2, the loss of model is further minimized, which denotes the model can further learn the ability of the MLLM-ISU task-related, *e.g.*, reasoning. Besides, in stage 3, the absolute changes of loss are the smallest. The main reason is that we use the well-designed VQA pairs to conduct refinement adjustments and learn fine-grained capabilities. Each of the three stages continuously learns the capability of coarse-grained understanding, reasoning understanding, and fine-grained understanding, respectively. These trends are reflected in the design of a three-stage progressive training strategy, and each stage plays a distinct and indispensable role.

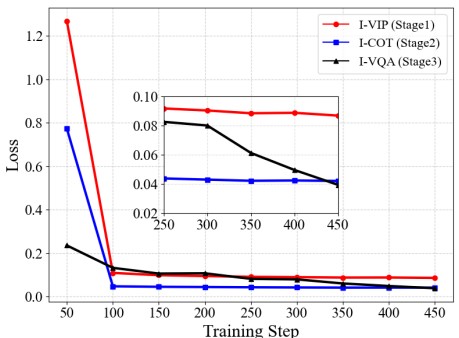

Figure 7: The training loss of three stages on Qwen2.5-VL-7B-Instruct model.

**Generalization verification.** We further go to verify the effectiveness of the three-stage training framework by the generalizability experiments. We use the trained model (3B and 7B) in normal weather to reason in adverse weather, *i.e.*, Normal→Foggy, as shown in Tab. 4. We can find that, 1) as the different stages of strategy are added, the understanding capability increases. 2) Compared with trained results in normal weather and the same setting, the results do not change much at each stage, even slightly above it, which indicates that the model has good generalization. More generalization experiments can be found in **Appendix F**.

Table 4: The results of generalization experiments.

| | | | | | | |
|---|---|---|---|---|---|---|
| | | Normal→Foggy | | | | |
| Model | Train stages | Avg. | Model | Train stages | Avg. | |
| | - | 50.88 | | - | 44.79 | |
| Qwen2.5-VL | w/ stage1 | 51.87 | Qwen2.5-VL | w/ stage1 | 61.27 | |
| -3B-Instruct [4] | w/ stage1&2 | 55.15 | -7B-Instruct [4] | w/ stage1&2 | 69.10 | |
| | w/ stage1&2&3 | **72.17** | | w/ stage1&2&3 | **77.66** | |

## 5.4 More Exploration Experiments and Analyses

**To what extent does model scale influence understanding performance?** In this subsection, we explore the relationship between the understanding performance of MLLMs and model scale. We use the InternVL2.5 series model to conduct the experiments due to its rich model scale, as shown in Fig. 8. We can find that, 1) as the model size continues to increase, the average comprehension (red line) increases, and the best performance can be reached when the model scale is 8B. Compared with the smallest InternVL2.5-1B model, the average performance can surpass it by 12.07%. However, some interesting phenomena can also be found in some subtasks, *e.g.*, the performance of InternVL2.5-2B

is lower than InternVL2.5-1B (-15.2%) in the IBJ subtask. In the IOL subtask, the best performance is the 4B model, not the 38B model. These findings indicate that we need to strike a balance between model scale and subtask performance to achieve better capability in intrusion scene understanding. More model scale results can be found in **Appendix G**.

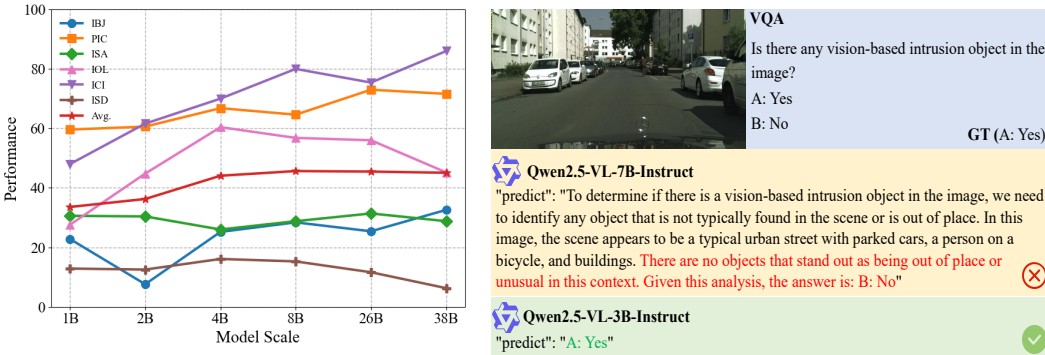

Figure 8: The performance of six model scales          Figure 9: Reasoning comparison

**Is a long chain of thought really good?** In previous experiments, we found that larger models do not perform as well as smaller models in the proposed MLLM-ISU task, *e.g.*, Qwen2.5-VL-7B-Instruct (42.05%)<Qwen2.5-VL-3B-Instruct (50.79%), DeepSeek-VL2-small (50.15%)<DeepSeek-VL2-tiny (49.47%). To explore the reasons, we compare the processes of the two models in a simple binary classification VQA pair, as shown in Fig. 9. We find that Qwen2.5-VL-7B-Instruct seems to think too much about the simple problems, which leads to poor or incorrect final results. The phenomenon suggests that long chain thinking is not always good. Besides, these results suggest that there are still some key issues that need to be addressed and improved in the MLLM-ISU task.

**How significant is the impact of training cost on performance.** In this question, we explore the impact of training cost on performance. Specifically, we set up five different training epochs (Epoch=15, 25, 35, 45, 50) to continuously expand the training cost of the model and use the Qwen2.5-VL-7B-Instruct to conduct the experiments, as shown in Fig. 10. We can find that the average performance does not continuously improve as training costs increase. It reaches its optimum at a certain critical point, *e.g.*, in stage 1, the best performance is reached at 65.20% in Epoch 35. Similarly, 69.00% in Epoch 35 (stage 2), 78.39% in Epoch 25 (stage 3). The main reason is that the SFT data is too small. As the training cost increases, the model can lead to overfitting and poor generalization performance. Therefore, we need to balance the relationship between the training data and the cost of the model to achieve the best performance.

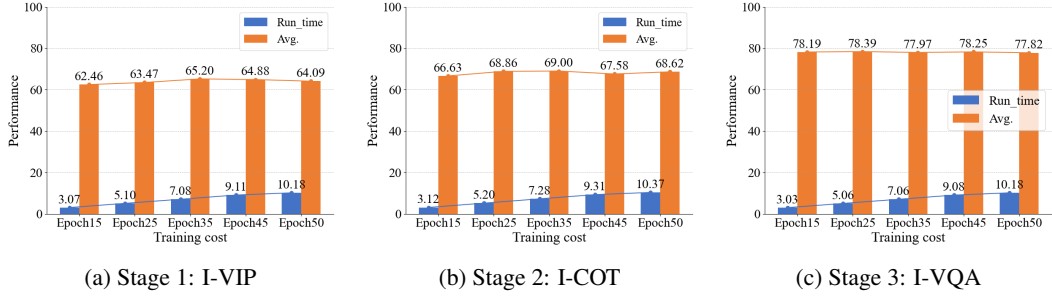

(a) Stage 1: I-VIP                    (b) Stage 2: I-COT                    (c) Stage 3: I-VQA

Figure 10: The performance impact of different training costs in three stages.

**Processing costs and latency.** We test the run-time, latency, and efficiency on two different models without any inference acceleration strategy, *i.e.*, 3B and 7B. The input size of the image is **1024×2048**, as shown in Tab. 5. We can find that our model has a low latency of 0.668 and a high predicted sample rate of 1.499.

Table 5: The experimental results of processing costs and latency on val (500 images).

| Model | Task | Runtime in val | Latency | Memory | Predicted_samples/s |
|---|---|---|---|---|---|
| 3B | IBJ | 5m34s | 0.668 | 9565MiB | 1.499 |
| 3B | PIC | 5m46s | 0.692 | 9565MiB | 1.446 |
| 7B | IBJ | 5m44s | 0.688 | 18441MiB | 1.453 |
| 7B | PIC | 5m47s | 0.694 | 18441MiB | 1.443 |

# 6 Discussions

**Dataset diversity.** To verify the universality of the proposed pipeline of VQA-Data Generation and enhance the diversity of intrusion scene types in real-world environments, we create a new benchmark dataset based on the BDD-100K for the MLLM-ISU task, namely MLLM-ISU-BDD. Due to space limitations, detailed information about the dataset can be found in **Appendix H**. Then, we also test the performance in some dominant MLLMs, as shown in Tab. 6. We can find that current MLLMs are not capable of intrusion scene understanding, especially in harsh and volatile scenarios. Besides, we can also find that even the best models still have a wide gap with human performance. These phenomena suggest that there is still some way to go for the task of intrusion scene understanding.

Table 6: Comprehensive performance evaluation of the 18 dominant MLLMs on the MLLM-ISU-BDD. The **bold** and underlined results denote the best and second-best performance.

| Model | Source | Release | IBJ | PIC | ISA | IOL | ICI | ISD | Average |
|---|---|---|---|---|---|---|---|---|---|
| Human | - | - | 90.00 | 96.67 | 38.89 | 85.56 | 96.67 | 35.61 | 73.90 |
| GPT-4o [19] | OpenAI | 2024-08 | 19.38 | 84.41 | 27.62 | 62.14 | 89.98 | 6.40 | 48.32 |
| LLaVa1.5-7B-hf [22] | UW-M&Micro | 2023-10 | 92.65 | 21.60 | 21.38 | 24.94 | 35.41 | 12.03 | 34.67 |
| LLaVa1.5-13B-hf [22] | UW-M&Micro | 2023-10 | 71.71 | 79.29 | 21.60 | 27.17 | 32.52 | 12.32 | 40.77 |
| Qwen2-VL-7B-Instruct [34] | Alibaba | 2024-06 | 25.84 | 86.19 | 36.53 | 50.56 | 59.91 | 6.07 | 44.18 |
| MiniCPM-V2.6 [37] | OpenBMB | 2024-08 | 5.79 | 83.96 | 30.29 | 33.85 | 77.95 | 12.44 | 40.71 |
| InternVL2.5-1B [5] | OpenGVLab | 2024-12 | 24.05 | 79.73 | 25.61 | 27.84 | 35.86 | 12.53 | 34.27 |
| InternVL2.5-2B [5] | OpenGVLab | 2024-12 | 5.35 | 83.96 | 30.73 | 45.66 | 43.88 | 11.69 | 36.88 |
| InternVL2.5-8B [5] | OpenGVLab | 2024-12 | 24.05 | 77.73 | 29.84 | 61.92 | 66.37 | 13.80 | 45.62 |
| DeepSeek-VL2-tiny [35] | DeepSeek | 2024-12 | 73.05 | 86.64 | 30.73 | 51.22 | 76.61 | 18.25 | **56.08** |
| DeepSeek-VL2-small [35] | DeepSeek | 2024-12 | 32.07 | 85.30 | 27.39 | 64.59 | 79.51 | 9.20 | 49.68 |
| Qwen2.5-VL-3B-Instruct [4] | Alibaba | 2025-01 | 35.41 | 77.73 | 29.18 | 89.76 | 80.62 | 5.79 | 53.08 |
| Qwen2.5-VL-7B-Instruct [4] | Alibaba | 2025-01 | 18.49 | 82.85 | 26.50 | 83.52 | 86.41 | 4.47 | 50.37 |
| Gemma3-4B-it [30] | Google | 2025-03 | 49.00 | 84.19 | 22.49 | 48.11 | 32.07 | 5.73 | 40.27 |
| Gemma3-12B-it [30] | Google | 2025-03 | 49.00 | 43.65 | 24.28 | 71.94 | 83.07 | 5.60 | 46.26 |
| Kimi-VL-A3B-Instruct [31] | Moonshot AI | 2025-04 | 16.93 | 86.86 | 26.50 | 36.08 | 88.42 | 17.21 | 45.33 |
| Kimi-VL-A3B-Thinking [31] | Moonshot AI | 2025-04 | 44.10 | 20.27 | 20.94 | 24.50 | 20.71 | 5.08 | 22.60 |
| InternVL3-2B [5] | OpenGVLab | 2025-04 | 18.04 | 75.72 | 33.18 | 42.32 | 77.28 | 6.57 | 42.19 |
| InternVL3-8B [5] | OpenGVLab | 2025-04 | 8.91 | 87.53 | 25.17 | 37.86 | 42.76 | 6.13 | 34.73 |

**Exploration of light task-specific strategy.** In Tab. 2, we adopt the default form to evaluate the performance of multiple MLLMs, *i.e.*, zero-shot manner. To understand upper-bound performance of MLLMs, we explore a light task-specific strategy, *e.g.*, prompt optimization, to understand upper-bound performance. We add a definitional prompt to facilitate better model understanding of the vision-based intrusion detection task, *i.e.*, <Vision-based intrusion detec-

Table 7: The quantitative results with the light task-specific strategy. Avg. denotes the average of six sub-tasks in different datasets.

| Model | Method | Avg. w/ MLLM-ISU-CS | Avg. w/ MLLM-ISU-BDD |
|---|---|---|---|
| Gemma3-4B-it [30] | Zero-shot | 42.16 | 40.27 |
| | w/ light strategy | 45.60 | 43.14 |
| | Performance Gain | +3.44 | +2.87 |
| Kimi-VL-A3B-Instruct [31] | Zero-shot | 42.14 | 45.33 |
| | w/ light strategy | 44.38 | 47.90 |
| | Performance Gain | +2.24 | +2.57 |
| InternVL3-2B [5] | Zero-shot | 42.82 | 42.19 |
| | w/ light strategy | 44.09 | 46.26 |
| | Performance Gain | +1.27 | +4.07 |

tion refers to judging whether a possible object exists in road>. Note that MLLM-ISU-CS and MLLM-ISU-BDD denote the benchmark datasets that are built based on the original Cityscape and BDD-100K datasets for our MLLM-ISU task, respectively. Besides, We test the light task-specific strategy on three MLLMs, *i.e.*, Gemma3-4B [30], Kimi-VL-A3B-Instruct [31], InternVL3-2B [5], as shown in Tab. 7. We can find that our strategy can improve the upper-bound performance, achieving average performance gains of 1.27 to 4.07, which proves the validity of our light task-specific strategy.

# 7 Conclusions

In this paper, we develop a new and vital task, Multimodal Large Language Models based Intrusion Scene Understanding (MLLM-ISU), to explore the capabilities of current MLLMs. To better accomplish the given task, we first design an effective automatic visual question-answer generation strategy and propose a novel MLLM-ISU dataset. Besides, we propose a three-stage post-training framework to improve understanding capability gradually. Finally, comprehensive experiments are conducted to evaluate the performance of current MLLMs and verify the effectiveness of the proposed framework. Our benchmark provides a new paradigm and direction for intrusion scene understanding tasks. In the future, we will explore effective strategies to improve the understanding capability of MLLMs.

## Acknowledgments and Disclosure of Funding

We sincerely thank the authors for their contributions. This work was fully supported by personal funding from the first author.

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

# Appendix for MLLM-ISU

## A   More VQA evaluation pairs cases in the MLLM-ISU dataset

In this subsection, we provide more VQA Paris cases. In every subtask, we report two different cases, as shown in Fig. 11. The VQA pairs contain six subtasks and are designed to evaluate the comprehensive and in-depth understanding capability of the current MLLMs. Intrusion Behavior Judgment and Person Intrusion Classification are the choice questions on binary classification. These two subtasks are used to test the capability of basic understanding and are relatively easy. Intrusion Summary Analysis, Intrusion Object Localization, and Intrusion Category Identification are used to test the capability of deeper levels of understanding and are relatively difficult. Intrusion Scene Descriptions is an open subtask and is designed to test the capability for open scene understanding. Our subtask is diverse and rich. The design VQA pairs can meet the requirements of the MLLM-ISU task and provide the foundation for the task.

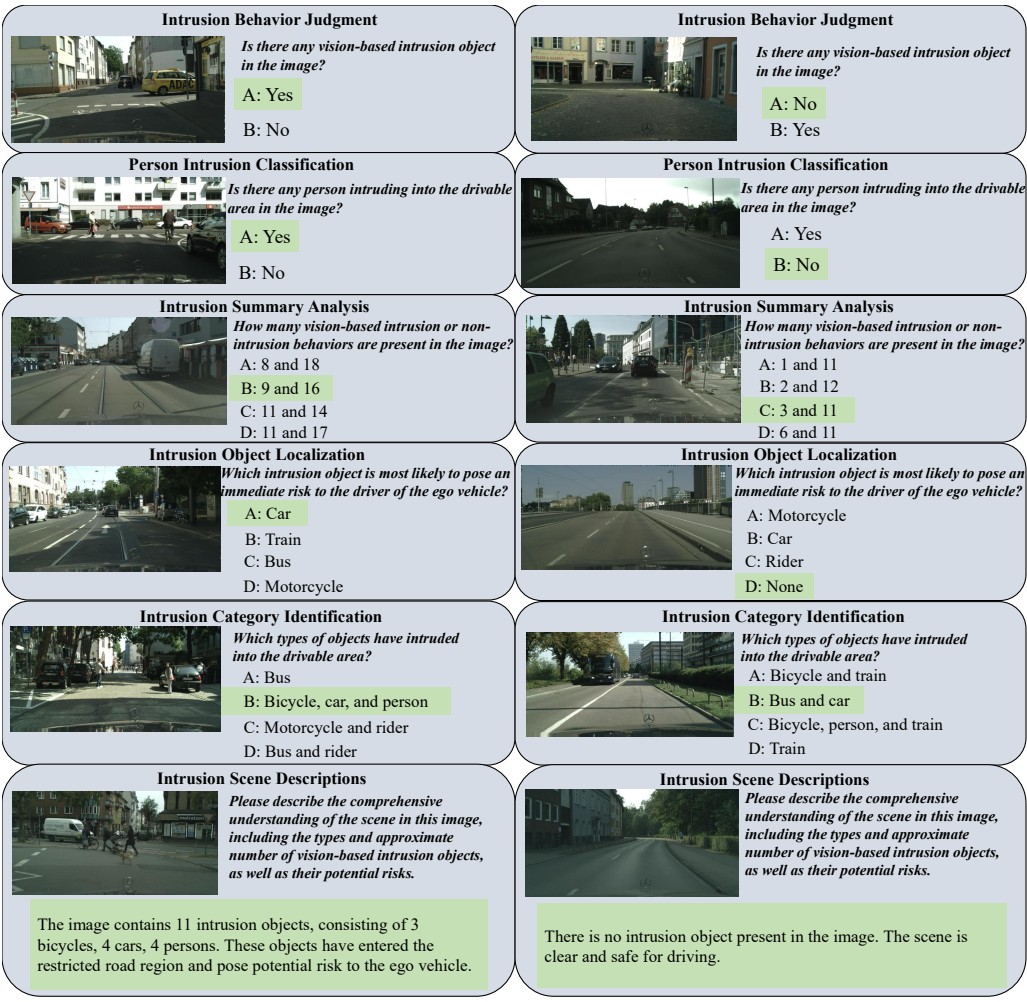

Figure 11: More VQA evaluation pairs cases in the MLLM-ISU dataset.

## B   Detailed prompt template of Intrusion-aware Visual Instruction Pre-training

In this subsection, we introduce the detailed prompt template for our Intrusion-aware Visual Instruction Pre-training strategy, as shown in Fig. 12. We show two different prompt template cases, intrusion

(Upper) and non-intrusion (Lower) cases, respectively. We use the same system prompt: *<Based on the visual information in the image, please describe any vision-based intrusion and non-intrusion behaviors in the scene.>*. Then, we give the corresponding answers. In this way, we construct the training dataset and perform Lora supervised fine-tuning to get the initial perception capabilities for MLLMs.

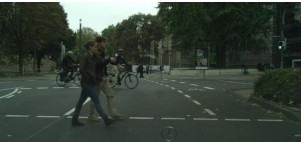

**<user>** Based on the visual information in the image, please describe any vision-based intrusion and non-intrusion behaviors in the scene.

**<assistant>** There are 9 intrusion objects in the image, including the following categories: 'bicycle, person, rider'. Besides, there are also 10 non-intrusion objects, including 'bus, person'. The object posing the greatest potential risk is a 'person', The main reason is that the category 'person' is closest to the driver of the ego vehicle and may pose a potential risk for the driver. Therefore, caution is advised, and the driver may need to take some necessary action.

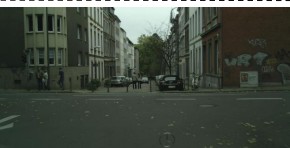

**<user>** Based on the visual information in the image, please describe any vision-based intrusion and non-intrusion behaviors in the scene.

**<assistant>** There is no vision intrusion object in the image. All 15 detected objects are non-intrusions, including the following categories: car, motorcycle, person. Therefore, the driver can safely drive.

Figure 12: Two different templates for Supervised Fine-Tuning in I-VIP (stage 1).

## C  More metrics and experimental results

In subsection **5.2**, for the subtask: Intrusion Scene Descriptions (ISD), we mainly adopt the **BLEU-4** to evaluate the performance. In this subsection, to calculate and provide a richer baseline, we also report some other metrics and performance for this subtask on some MLLMs, *i.e.*, Rouge-1, Rouge-2, Rouge-L, as shown in Tab. 8. Note that MLLM-ISU-CS and MLLM-ISU-BDD denote the benchmark datasets that are built based on Cityscape and BDD-100K datasets for our MLLM-ISU task, respectively.

Table 8: Some other metrics results for the ISD subtask.

| Model | Source | Release | MLLM-ISU-CS | | | MLLM-ISU-BDD | | |
|---|---|---|---|---|---|---|---|---|
| | | | Rouge-1 | Rouge-2 | Rouge-L | Rouge-1 | Rouge-2 | Rouge-L |
| GPT-4o[19] | OpenAI | 2024-08 | 15.75 | 1.80 | 7.50 | 14.78 | 1.62 | 7.03 |
| LLaVa1.5-13B-hf[22] | UW-M&Micro | 2023-10 | 19.95 | 3.33 | 12.29 | 20.64 | 3.39 | 12.32 |
| MiniCPM-V2.6 [37] | OpenBMB | 2024-08 | 19.14 | 2.21 | 11.42 | 19.86 | 2.48 | 12.07 |
| InternVL2.5-2B[5] | OpenGVLab | 2024-12 | 18.73 | 2.23 | 11.54 | 18.63 | 2.30 | 10.84 |
| InternVL2.5-8B[5] | OpenGVLab | 2024-12 | 21.49 | 2.34 | 13.40 | 20.10 | 1.96 | 12.07 |
| DeepSeek-VL2-tiny[35] | DeepSeek | 2024-12 | 22.66 | 4.31 | 16.49 | 23.02 | 4.43 | 16.08 |
| DeepSeek-VL2-small[35] | DeepSeek | 2024-12 | 20.41 | 2.32 | 8.60 | 19.09 | 1.65 | 8.98 |
| Qwen2.5-VL-3B-Instruct[4] | Alibaba | 2025-01 | 16.44 | 1.72 | 6.94 | 14.61 | 1.53 | 6.19 |
| Qwen2.5-VL-7B-Instruct[4] | Alibaba | 2025-01 | 14.74 | 1.64 | 5.07 | 13.75 | 1.57 | 4.86 |
| Gemma3-4B-it[30] | Google | 2025-03 | 14.56 | 1.92 | 5.31 | 13.82 | 1.74 | 5.07 |
| Gemma3-12B-it[30] | Google | 2025-03 | 13.77 | 1.45 | 5.40 | 13.73 | 1.53 | 5.23 |
| Kimi-VL-A3B-Instruct[31] | Moonshot AI | 2025-04 | 21.50 | 2.90 | 11.30 | 24.04 | 3.16 | 15.07 |
| Kimi-VL-A3B-Thinking[31] | Moonshot AI | 2025-04 | 13.98 | 1.61 | 5.41 | 13.43 | 1.72 | 5.46 |
| InternVL3-2B[5] | OpenGVLab | 2025-04 | 15.75 | 1.44 | 6.67 | 15.42 | 1.49 | 6.63 |
| InternVL3-8B[5] | OpenGVLab | 2025-04 | 16.53 | 1.64 | 6.67 | 15.42 | 1.65 | 6.21 |

## D  More training details

In this appendix, we present more details of the training for the proposed three-stage framework. For the cutoff_len parameters, we use 2048. For the learning_rate, we adopt the default setting, *i.e.*, 5e-5. All stages adopt the Lora as a supervised fine-tuning method.

Table 9: The detail setting for the training experiments.

| Setting | Value | Setting | Value |
|---|---|---|---|
| cutoff_len (stage1) | 2048 | preprocessing_num_workers | 16 |
| cutoff_len (stage2) | 2048 | per_device_train_batch_size | 1 |
| cutoff_len (stage3) | 2048 | per_device_eval_batch_size | 1 |
| gradient_accumulation_steps | 8 | learning_rate | 5e-5 |
| num_train_epochs (3B/7B) | 2/5 | finetuning_type | lora |

## E  More metrics and results of three post-training stages

We further verify the effectiveness of the proposed three-Stage Post-Training Framework. Like the previous experiment, we use the Qwen2.5-VL-7B-Instruct to conduct the experiment in five different epochs, *i.e.*, Epoch=15, 25, 35, 45, 50, as shown in Tab. 10. We can find that as the different stages are added, the performance increases and reaches 78.19%, 78.39%, 77.97%, 78.25%, and 77.82%, respectively. Besides, in different subtasks, our framework can also give a performance gain, which denotes that the three different Supervised Fine-tuning strategies are effective, *i.e.*, Perception (Intrusion-aware Visual Instruction Pre-training)→Reasoning (Intrusion Chain of Thought Tuning)→Understanding (Intrusion-centric VQA Tuning). We also give the training loss in different epochs, as shown in Fig. 13. We can find that in different epochs, models can learn the different capabilities of the three stages. As the training step increases, the loss of the model changes less, especially after 1500 steps. Therefore, we believe it is important to choose appropriate training steps.

Table 10: More performance results of the proposed three post-training stages on different MLLMs. I-VIP, I-COT, and I-VQA denote the proposed three different strategies in the training stages.

| Model+Method | IBJ | PIC | ISA | IOL | ICI | ISD | Average |
|---|---|---|---|---|---|---|---|
| *7B Open-source MLLMs, Epoch=15* | | | | | | | |
| Qwen2.5-VL-7B-Instruct [4] | 16.00 | 61.40 | 24.00 | 64.80 | 81.40 | 4.68 | 42.05 |
| Qwen2.5-VL-7B-Instruct+I-VIP | 81.00 | 65.60 | 24.40 | 81.80 | 92.00 | 29.93 | 62.46 |
| Qwen2.5-VL-7B-Instruct+I-VIP+I-COT | 95.40 | 68.40 | 22.80 | 81.60 | 96.60 | 34.97 | 66.63 |
| Qwen2.5-VL-7B-Instruct+I-VIP+I-COT+I-VQA | 97.00 | 79.40 | 48.80 | 93.20 | 99.40 | 51.33 | **78.19** |
| *7B Open-source MLLMs, Epoch=25* | | | | | | | |
| Qwen2.5-VL-7B-Instruct [4] | 16.00 | 61.40 | 24.00 | 64.80 | 81.40 | 4.68 | 42.05 |
| Qwen2.5-VL-7B-Instruct+I-VIP | 80.80 | 65.80 | 27.40 | 83.00 | 94.00 | 29.84 | 63.47 |
| Qwen2.5-VL-7B-Instruct+I-VIP+I-COT | 92.60 | 72.40 | 29.40 | 84.00 | 95.40 | 39.35 | 68.86 |
| Qwen2.5-VL-7B-Instruct+I-VIP+I-COT+I-VQA | 95.20 | 77.80 | 54.00 | 93.40 | 99.20 | 50.73 | **78.39** |
| *7B Open-source MLLMs, Epoch=35* | | | | | | | |
| Qwen2.5-VL-7B-Instruct [4] | 16.00 | 61.40 | 24.00 | 64.80 | 81.40 | 4.68 | 42.05 |
| Qwen2.5-VL-7B-Instruct+I-VIP | 87.60 | 67.40 | 29.80 | 82.40 | 94.20 | 29.79 | 65.20 |
| Qwen2.5-VL-7B-Instruct+I-VIP+I-COT | 94.20 | 69.20 | 31.80 | 83.80 | 96.40 | 38.58 | 69.00 |
| Qwen2.5-VL-7B-Instruct+I-VIP+I-COT+I-VQA | 96.40 | 77.60 | 52.00 | 92.00 | 99.20 | 50.61 | **77.97** |
| *7B Open-source MLLMs, Epoch=45* | | | | | | | |
| Qwen2.5-VL-7B-Instruct [4] | 16.00 | 61.40 | 24.00 | 64.80 | 81.40 | 4.68 | 42.05 |
| Qwen2.5-VL-7B-Instruct+I-VIP | 84.40 | 66.60 | 27.60 | 85.80 | 95.00 | 29.90 | 64.88 |
| Qwen2.5-VL-7B-Instruct+I-VIP+I-COT | 94.00 | 66.60 | 27.60 | 84.60 | 96.80 | 35.89 | 67.58 |
| Qwen2.5-VL-7B-Instruct+I-VIP+I-COT+I-VQA | 96.20 | 78.40 | 51.00 | 93.40 | 99.20 | 51.28 | **78.25** |
| *7B Open-source MLLMs, Epoch=50* | | | | | | | |
| Qwen2.5-VL-7B-Instruct [4] | 16.00 | 61.40 | 24.00 | 64.80 | 81.40 | 4.68 | 42.05 |
| Qwen2.5-VL-7B-Instruct+I-VIP | 84.80 | 66.40 | 27.60 | 81.80 | 94.00 | 29.94 | 64.09 |
| Qwen2.5-VL-7B-Instruct+I-VIP+I-COT | 93.80 | 66.20 | 31.80 | 83.60 | 95.40 | 40.89 | 68.62 |
| Qwen2.5-VL-7B-Instruct+I-VIP+I-COT+I-VQA | 95.60 | 78.00 | 52.60 | 91.60 | 98.80 | 50.29 | **77.82** |

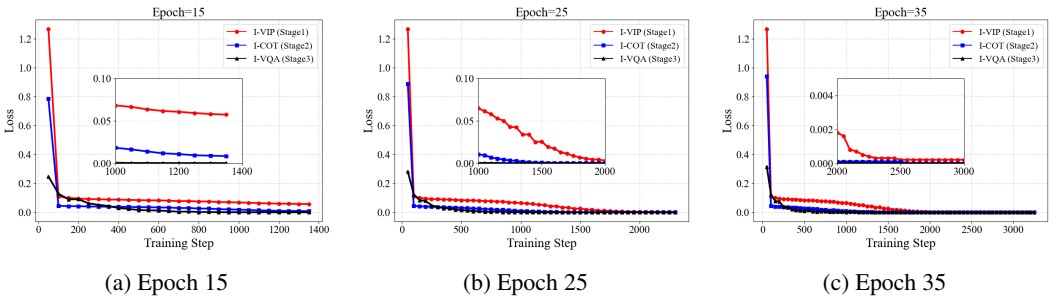

(a) Epoch 15                                (b) Epoch 25                                (c) Epoch 35

Figure 13: The training loss in three different epochs.

# F   More generalization verification experiments

In this appendix, we further conduct more generalization verification experiments and report more experimental results for the three-stage training framework. Specifically, we use the three different foggy coefficients to conduct, *i.e.*, $\alpha$=0.02, $\alpha$=0.01, and $\alpha$=0.005, respectively. The Qwen2.5-VL-3B-Instruct [4] and Qwen2.5-VL-7B-Instruct [4] are chosen as the baseline model, and the result is shown in Tab. 11. We can find that our three-stage training framework has strong generalization performance and shows promising performance on several different tasks.

Table 11: More generalization results of the proposed three post-training stages on different tasks. Note that our strategy is to increase them one by one. $\alpha$ denotes the foggy coefficients in Cityscape [7].

| Model | Train stages | IBJ | PIC | ISA | IOL | ICI | ISD | Avg. |
|---|---|---|---|---|---|---|---|---|
| Normal→Foggy, $\alpha = 0.02$ | | | | | | | | |
| Qwen2.5-VL -3B-Instruct [4] | - | 47.00 | 61.20 | 31.00 | 78.00 | 81.60 | 6.51 | 50.88 |
| | w/ stage1 | 51.60 | 61.60 | 28.60 | 78.80 | 83.80 | 6.81 | 51.87 |
| | w/ stage1&2 | 55.80 | 57.60 | 25.00 | 81.60 | 85.60 | 25.30 | 55.15 |
| | w/ stage1&2&3 | 94.40 | 67.80 | 37.40 | 87.40 | 97.60 | 48.41 | **72.17** |
| Normal→Foggy, $\alpha = 0.01$ | | | | | | | | |
| Qwen2.5-VL -3B-Instruct [4] | - | 45.80 | 62.20 | 30.60 | 78.20 | 82.40 | 6.58 | 50.96 |
| | w/ stage1 | 50.80 | 61.40 | 28.00 | 80.00 | 84.40 | 6.67 | 51.88 |
| | w/ stage1&2 | 55.20 | 58.20 | 24.60 | 81.20 | 87.40 | 25.02 | 55.27 |
| | w/ stage1&2&3 | 94.40 | 68.20 | 36.60 | 88.40 | 97.40 | 48.22 | **72.20** |
| Normal→Foggy, $\alpha = 0.005$ | | | | | | | | |
| Qwen2.5-VL -3B-Instruct [4] | - | 45.00 | 60.80 | 30.40 | 78.40 | 82.40 | 6.54 | 50.59 |
| | w/ stage1 | 50.80 | 61.80 | 27.80 | 81.00 | 84.00 | 6.73 | 52.02 |
| | w/ stage1&2 | 55.00 | 58.00 | 24.60 | 83.80 | 88.00 | 25.42 | 55.80 |
| | w/ stage1&2&3 | 94.40 | 67.60 | 38.20 | 88.00 | 97.60 | 48.34 | **72.36** |
| Normal→Foggy, $\alpha = 0.02$ | | | | | | | | |
| Qwen2.5-VL -7B-Instruct [4] | - | 31.20 | 62.60 | 24.20 | 63.60 | 82.40 | 4.72 | 44.79 |
| | w/ stage1 | 76.40 | 63.40 | 23.80 | 80.80 | 92.80 | 30.41 | 61.27 |
| | w/ stage1&2 | 94.00 | 63.00 | 24.20 | 82.60 | 96.00 | 54.81 | 69.10 |
| | w/ stage1&2&3 | 95.20 | 77.40 | 49.80 | 94.00 | 99.20 | 50.33 | **77.66** |
| Normal→Foggy, $\alpha = 0.01$ | | | | | | | | |
| Qwen2.5-VL -7B-Instruct [4] | - | 26.60 | 63.00 | 24.40 | 61.00 | 82.00 | 4.75 | 43.63 |
| | w/ stage1 | 81.00 | 63.60 | 23.80 | 80.60 | 92.00 | 30.41 | 61.90 |
| | w/ stage1&2 | 94.40 | 64.20 | 24.80 | 81.60 | 96.00 | 55.24 | 69.37 |
| | w/ stage1&2&3 | 95.60 | 79.20 | 51.20 | 94.20 | 99.20 | 50.34 | **78.29** |
| Normal→Foggy, $\alpha = 0.005$ | | | | | | | | |
| Qwen2.5-VL -7B-Instruct [4] | - | 22.20 | 62.40 | 24.60 | 62.00 | 81.00 | 4.72 | 42.82 |
| | w/ stage1 | 85.40 | 63.60 | 24.00 | 79.60 | 91.40 | 30.67 | 62.45 |
| | w/ stage1&2 | 94.60 | 65.40 | 24.60 | 82.20 | 96.00 | 55.79 | 69.77 |
| | w/ stage1&2&3 | 96.20 | 79.60 | 52.80 | 93.60 | 99.20 | 50.87 | **78.71** |

## G   More model scale results on InternVL3-series models

In addition to InternVL2.5-series models, we also conduct model scale experiments in the latest InternVL3-series model, as shown in Fig. 14. We can find that, like the InternVL2.5-series model, the best average performance can be reached when the model scale is 9B, not the largest 38B model. We think this phenomenon has something to do with overthinking the model, where overthinking simple problems instead creates illusions. This is something we need to study further.

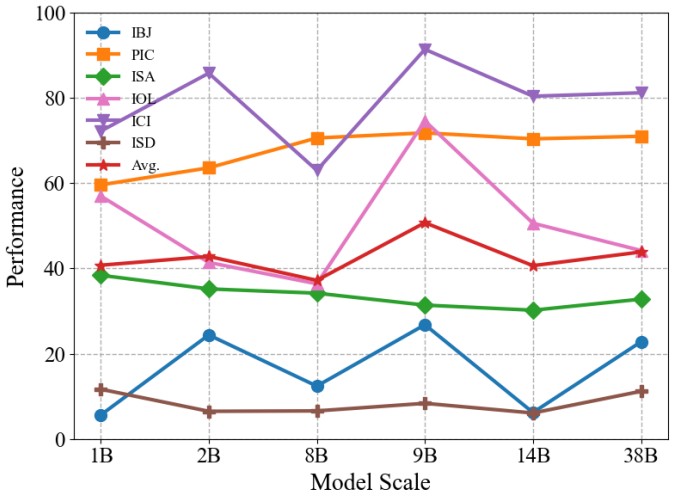

Figure 14: The model scale results on InternVL3-series

## H   The detailed information for proposed MLLM-ISU-BDD

To verify the universality of the proposed pipeline of VQA-Data Generation and enhance the diversity of intrusion scene types in real-world environments, we create a new benchmark dataset for the MLLM-ISU task, namely MLLM-ISU-BDD. The MLLM-ISU-BDD is built based on the BDD-100K datasets. The detailed method can refer to Fig. 3. Our new MLLM-ISU-BDD datasets contain rich intrusion scene types, *e.g.*, multiple different weather (Clear, Cloudy, Rainy, Foggy, Night), different geographic environment (City, Highway, Suburban/Rural), different period of time (Daytime, Dusk, Night), and Different transportation environments (Heavy Traffic, Empty Road). We clean the original dataset based on the proposed intrusion detection task features. Finally, our datasets contain 8892 training Pairs and 2694 VQA evaluation Pairs. Our extended benchmark explicitly includes nighttime, adverse weather, and non-urban roads, enabling more comprehensive evaluation of the intrusion scene understanding task in real-world environments.

## I   More discussion and interesting finding

**Discussion on model version performance variations.** In Tab. 2 and Tab. 6, we observed that newer versions of multimodal large models (MLLMs) do not always outperform their predecessors on our proposed intrusion scene understanding task, *e.g.*, in Tab. 2, InternVL3-8B is lower than InternVL2.5-8B, the interesting phenomenon also occurs in previous work [18]. We think this is reasonable. The main reason is that, during the model update process, priority is typically given to improving abstract reasoning, instruction-following, and general linguistic capabilities rather than low-level visual perception. Consequently, newer models may excel in complex reasoning and multi-turn understanding but show reduced sensitivity to small, partially occluded, or contextually subtle intrusion targets. Besides, stronger reasoning abilities typically imply longer chains of thought. However, in section 5.4, we find that longer reasoning isn't always effective for our tasks. These findings indicate that when applying general MLLMs to visual intrusion understanding, we need to explore the task-specific adaptation strategies to enhance their perception and recognition of fine-grained intrusion cues.

## J Limitations

To the best of our knowledge, the MLLM-ISU is proposed for the first time and is the first attempt in the intrusion detection field. We believe our work will produce positive effects in several application areas, *e.g.*, autonomous driving, intelligent monitoring, and security. We believe designing more comprehensive benchmarks, *e.g.*, richer understanding tasks, and exploring more efficient improvement training or training-free strategies, *e.g.*, training-free reasoning method (Retrieving Augmented Generation), is a worthy research direction in the future.

