# OpenReview forum: "MLLM-ISU: The First-Ever Comprehensive Benchmark for Multimodal Large Language Models based Intrusion Scene Understanding"
_NeurIPS.cc/2025/Datasets_and_Benchmarks_Track — NeurIPS 2025 Datasets and Benchmarks Track poster_

### Official Review · Reviewer_ePVt · 2025-06-30

**Rating:** 4
**Confidence:** 3

**Summary:**

This paper proposes Multimodal Large Language Models based Intrusion Scene Understanding (MLLM-ISU). An automated visual question-answering generation strategy is designed and a new MLLM-ISU dataset is constructed. A post-training framework is proposed, which is divided into three consecutive training stages: intrusion-aware visual instruction pre-training, intrusion thought chain tuning, and intrusion-centric visual question-answering tuning. Experiments verify the effectiveness of the method.

**Additional Feedback:**

As the Limitations.

**Dataset Code Accessibility:**

No

**Dataset Code Comments:**

No code or dataset links are provided

**Ethical Considerations:**

No, there are no or only very minor ethics concerns

**Final Justification:**

I appreciate the author's response. The more detailed response explains the paper's claims and its advantages compared to the previous intrusion detection methods. This clarifies my questions, so I keep my original rating.

**Limitations Weaknesses:**

1. In Section 3.3, line 150, "Compared to some automatic Question-Answer generation methods by using MLLMs (GPT-4V or GPT-4o), e.g., MMAD, FAVOR-Bench, Biology Instructions, our proposed strategy presents more efficiency and accuracy, as shown in Fig. 3". However, there is no relevant experiment to prove that "The proposed strategy presents more efficiency and accuracy". Figure 3 does not further explain this statement.

2. In Section 5.4, the explanation of the experimental phenomenon is unclear. Line 319, "The main reason is that the SFT data is too small." This is ambiguous. What does SFT data represent? Is it the proposed dataset or other datasets? Please clarify and explain.

3. On a single task, does the proposed method have better performance than the previous intrusion detection methods MF-ID, MMID and Ada-iD?

**Strengths Contributions:**

1. Intrusion scene understanding is a novel and interesting research.

2. MLLM-ISU can understand multi-tasks and is worth exploring.

---

> ### Author Rebuttal · Authors · 2025-07-31
>
> **Q1: In Section 3.3, line 150, "Compared to some automatic Question-Answer generation methods by using MLLMs (GPT-4V or GPT-4o), e.g., MMAD, FAVOR-Bench, Biology Instructions, our proposed strategy presents more efficiency and accuracy, as shown in Fig. 3". However, there is no relevant experiment to prove that "The proposed strategy presents more efficiency and accuracy". Figure 3 does not further explain this statement.**
>
> **A1**: Thanks! We clarify that our sample generation method differs fundamentally from GPT-4o. We adopt a **Python structured template-driven approach** method. Specifically, our dataset is programmatically constructed via Python scripts using controlled templates, which ensures consistent formatting, high data throughput, and minimal post-processing. In contrast, existing methods often rely on MLLMs (e.g., GPT-4V) to auto-generate questions, which typically require human validation and filtering to ensure correctness and relevance. Each VQA Pair is generated from well-designed task-specific templates. Therefore, our strategy leads to significantly reduced human involvement and more deterministic generation, contributing to both higher efficiency and better factual correctness. Here, we also give some case analysis below.
>
> **Case analysis:**
>
> **Method: GPT-4o**
>
> Prompt:
>
> You are an autonomous driving safety analyst. Your task is to generate a **question-answer pair** for vision-based intrusion detection based on the input image. Vision-based intrusion refers to judging whether a possible object exists in the restricted area-of-interest, e.g., pedestrian, bicycle, or car. Please understand the given image and generate an Image-caption Pair. After generating the Image-caption Pair, please generate a VQA sample pair according to our following task requirements. Only generate the caption and the VQA block. Do not include other text or explanation.
>
> **Task 1**: Please generate a VQA for determining whether there is an intrusion or non-intrusion in this image.
>
> **Task 2**: Please generate a VQA for determining whether there is person intrusion or non-intrusion behavior in this image.
>
> **GPT4o answer**: (**5.56s**)
>
> Image Caption:
>
> A cyclist is entering the roadway from the sidewalk on the right side of a roundabout, near two moving vehicles and road signs indicating slippery conditions.
>
> Task 1 – Intrusion VQA
>
> Q: Is there any vision-based intrusion object in the image?
>
> A: Yes, a cyclist is entering the restricted road area.
>
> Task 2 – Person Intrusion VQA
>
> Q: Is there any person intrusion behavior in the image?
>
> A: Yes, a person with a bicycle is intruding into the road area.
>
>
> **Ours method and answer:**
>
> Is there any vision-based intrusion object in the image? \nA: Yes \nB: No
>
> Answer: "A"
>
> Is there any person intruding into the drivable area in the image?  \nA: Yes \nB: No
>
> Answer: "No"
>
>
> | Method                                   | Design Prompts | Question Type | Labor Cost | Generation Speed                     | Reliability                         |
> | ---------------------------------------- | :--------------: | -------- | ---------- | ------------------------------------ | ----------------------------- |
> | GPT-4o                                   | √              | Complex  | Much      | Slow (a few seconds to tens of seconds) | Maybe hallucination  |
> | Ours (Python structured template-driven) | ×              | Simple   | Little     | Fast (milliseconds)                  | Based on GT labeling, No hallucinate |
>
>
>
> **Q2: In Section 5.4, the explanation of the experimental phenomenon is unclear. Line 319, "The main reason is that the SFT data is too small." This is ambiguous. What does SFT data represent? Is it the proposed dataset or other datasets? Please clarify and explain.**
>
> **A2**: Thanks! We clarify and explain the "The main reason is that the SFT data is too small." The SFT datasets denotes our proposed MLLM-ISU datasets. We will modify it in the final revision.
>
>
>
> **Q3: On a single task, does the proposed method have better performance than the previous intrusion detection methods MF-ID, MMID and Ada-iD?**
>
> **A3**: Thanks! The brand new MLLM-ISU task we propose is fundamentally different from some previous work, as shown in the table below. Our MLLM-ISU task aims to explore a comprehensive benchmark for the visual intrusion detection task. Our work focuses on generalization and zero-shot performance, whereas these approaches emphasize task-specific design and fine-tuning. More importantly, we primarily focus on the intrusion scene understanding capability; however, this capability is not available in MF-ID, MMID, and Ada-iD models. We will subsequently explore reinforcement learning-based approaches, i.e., Group Relative Policy Optimization (GRPO), to fine-tune the performance of the model for detection, segmentation and intrusion detection.
>
>
> | Intrusion tasks       | Task Type                  | Structure | Modal         | Zero-shot | Scene Understanding |
> | --------------------- | -------------------------- | --------- | ------------- | :---------: | -------------------- |
> | MF-ID [10]            | Detectable Single Task     | Close     | Single-Modal  | ×         | low                  |
> | MM-ID [11]            | Detectable Single Task     | Close     | Single-Modal  | ×         | low                   |
> | Ada-iD [9]            | Detectable Single Task     | Close     | Single-Modal  | ×         | low                  |
> | MLLM-ISU              | Comprehension Multi Task   | Open      | Multi-Modal   | √         | high                   |

---

> > ### Comment · Reviewer_ePVt · 2025-08-05
> >
> > The answer to Q1 doesn't address my question. The authors claim that "The proposed strategy presents more efficiency and accuracy." However, in the case study they provide, Q: Is there any person intrusion behavior in the image? The proposed method's answer seems misleading compared to GPT4o's. If there's a cyclist entering the roadway in the image, that also implies a person intruding into the drivable area. The model output doesn't match the paper's claims.
> >
> > The answer to Q3 makes me think the proposed benchmark has limited application scenarios.

---

> > > ### Author Response · Authors · 2025-08-05
> > > **Response to the reviewer's question**
> > >
> > > We sincerely thank the reviewer for the constructive feedback. Below we provide our detailed response.
> > >
> > > **Q1: The answer to Q1...... The model output doesn't match the paper's claims.**
> > >
> > > **A1.1:** First of all, let me re-explain the principle of our proposed strategy.
> > >
> > > **(1) Accuracy.** We first carefully designed an intrusion detection program to obtain preliminary intrusion and no-intrusion labels. For the intrusion detection program, we refer to previous methods [10]. Then, these pre-determined intrusion and non-intrusion labels are written into the original XML files. Note that the original XML files contain information about the bounding box and categories. This ensures the accuracy of both intrusion and non-intrusion in our final input content. Compared to the GPT4o method, which can be hallucinatory (since GPT4o needs first to understand the image, then determine intrusions and non-intrusions), our methods of intrusions and non-intrusions are more accurate.
> > >
> > > **(2) Generation speed.** Since we have the accuracy intrusion and non-intrusion labels and the number and names of categories in the XML, we can use Python to design structured question prompts and answers to generate VQA pairs. This gives our approach a much faster generation speed relative to GPT4o.
> > >
> > > **A1.2:** Besides, for your new feedback. We will clarify this from the following aspects.
> > >
> > > **(1) Task Definition Based on Cityscapes Taxonomy.** Our benchmark construction is based on the Cityscapes dataset, which explicitly distinguishes between the “person” and “rider” categories. In this taxonomy:
> > >
> > > Person: refers to pedestrians or standing individuals,
> > >
> > > Rider: refers to people actively riding bicycles, motorcycles, etc.
> > >
> > > In our “Person Intrusion Classification (PIC)” sub-task, we strictly follow this category-level separation. Hence, a cyclist is treated as a “rider”, not a “person”, and does not count as a “person” intrusion under our benchmark’s task definition.
> > >
> > > **(2) Fine-grained evaluation.** As described in our paper: “The subtask of PIC is designed to test the model’s intrusion judgments for individual categories. It is used to measure the model’s ability to categorize intrusion events at a fine-grained level.” For this sub-task, we refer to previous works [10] that demonstrate intrusion detection as a fine-grained identification task. Besides, [10] denotes that the “person” and “rider” have similar features, e.g., shape features, edge features. Therefore, we follow previous works and distinguish between “person” and “rider” to test the fine-grained comprehension of the model, which is consistent with the purpose of our original intrusion scene understanding task.
> > >
> > > **(3) Additional Clarification Will Be Added.** We will clarify this semantic distinction in the manuscript and explicitly define what counts as “person intrusion” vs. “rider intrusion” under our benchmark setup.
> > >
> > > **Q2: The answer to Q3... limited application scenarios.**
> > >
> > > **A2:** Compared with previous intrusion detection works, our work has diverse application scenarios.
> > >
> > > **(1) Rich intrusion categories.** Previous works have limited intrusion categories (1 or 4), our work includes more common categories of intrusion (8). Besides, these works focus on image understanding of structured outputs,e.g., object detection and semantic segmentation, and lack the capabilities of visual reasoning guided by linguistic instructions.
> > >
> > > **(2) More complex or real-world environments.**  To enhance the diversity of intrusion scene types and improve the generalizability in real-world environments, we create a new benchmark dataset for the MMLM-ISU task, namely MMLM-ISU-BDD. The MMLM-ISU-BDD is built based on the BDD-100K datasets. Our new MMLM-ISU-BDD datasets contain rich intrusion scene types, e.g., various weather conditions, geographic environments, periods, and transportation environments. We also conduct comprehensive experiments on multiple MLLMs and find that current MLLMs are not capable of intrusion scene understanding, especially in harsh and volatile scenarios. Besides, we also find that even the best models (56.08) still have a wide gap with human performance (73.90).
> > >
> > > **(3) General-purpose vs. Specialized model.** We are the first to propose a three-stage post-training framework for MLLMs in the intrusion scene understanding task. Unlike prior task-specific methods, our approach emphasizes general-purpose — it supports zero-shot, multi-class, and open-ended intrusion reasoning. While it may not outperform specialized models on a single task, it shows comprehensive versatility across diverse intrusion scenarios.
> > >
> > > **(4) Multi-Dimensional Contributions.** Our contributions are comprehensive, not limited to model performance.
> > > We propose a new task, a new benchmark, comprehensive evaluation, and a new post-training method, all designed for MLLM-based intrusion scene understanding task. Our benchmark can serve as an foundation for further
> > > research in intrusion detection field.

---

> > > > ### Comment · Reviewer_ePVt · 2025-08-06
> > > >
> > > > I appreciate the author's response. The more detailed response explains the paper's claims and its advantages compared to the previous intrusion detection methods. This clarifies my question, so I will keep my original rating.

---

### Official Review · Reviewer_D4oT · 2025-07-02

**Rating:** 4
**Confidence:** 4

**Summary:**

This paper presents MLLM-ISU, a novel benchmark and training framework for evaluating and improving Multi-modal Large Language Models (MLLMs) on visual intrusion understanding tasks. The authors construct a new dataset from Cityscapes with multi-task annotations and design a three-stage supervised fine-tuning strategy (I-VIP, I-COT, I-VQA) to enhance model performance across various sub-tasks. Extensive experiments show that the proposed benchmark and training pipeline significantly improve MLLM performance and offer valuable insights into model capability, generalization, and training dynamics.

**Additional Feedback:**

1. How well does the model trained on Cityscapes generalize to other intrusion scenarios in rural, indoor, or low-light environments? Could additional diverse datasets be included to enhance robustness?
2. Could the authors discuss the trade-off between training cost and performance gain more precisely? Is there an optimal configuration (e.g., epoch count or stage dropout)?
3. Does the introduction of long-chain reasoning (I-COT stage) consistently benefit all tasks, or are there cases where it induces overthinking and misclassification (as hinted in Section 5.4)?
4. Could the authors consider incorporating human feedback (RLHF) or uncertainty calibration in future extensions?

**Dataset Code Accessibility:**

NA; not applicable to this submission (e.g., no new dataset, benchmark, code, or data provided)

**Dataset Code Comments:**

No.
Please kindly provide the code and the datasets.

**Ethical Considerations:**

No, there are no or only very minor ethics concerns

**Limitations Weaknesses:**

1. Limited Comparison with Prior Benchmarks: While the proposed benchmark is comprehensive, comparisons with other visual reasoning or security-related datasets are lacking, which limits understanding of relative difficulty and scope.
2. Data Bias Risk: Since the entire benchmark is based on Cityscapes, the generalizability to non-urban scenes or real-world surveillance data might be limited.
3. Model Scale: Some observations (e.g., smaller models outperforming larger ones) are reported but not deeply analyzed, and no concrete recommendation is given.
4. Insufficient analysis of failure cases: While Section 5 illustrates that models may exhibit "overthinking" during multi-turn reasoning, it lacks an in-depth analysis of the failure causes. For instance, is the failure due to inadequate visual grounding during I-VIP, reasoning drift in I-COT, or hallucinated answers in I-VQA? A more systematic diagnosis would help clarify the limitations of each stage.
5. Performance evaluation metrics: The paper lacks detailed explanation for the chosen evaluation metrics. Please provide a more comprehensive description of each metric used, including its relevance to the task. Additionally, are there alternative metrics that may be more suitable for evaluating these tasks, such as task-specific accuracy, human preference scores, or semantics-aware metrics?

**Strengths Contributions:**

1. Comprehensive Benchmark: The proposed benchmark covers six intrusion-related tasks with both classification and open-ended formats, enabling holistic evaluation of MLLMs on visual understanding.
2. Systematic Training Pipeline: The three-stage fine-tuning strategy is well-designed and progressively improves model capabilities from coarse-level perception to fine-grained reasoning.
3. Detailed Experimental Analysis: The authors provide rich ablation studies, generalization analysis (Normal → Foggy), and stage-wise impact validation.
4. Novel Task Setting: Introducing structured intrusion understanding tasks for MLLMs is a valuable and original contribution.

---

> ### Author Rebuttal · Authors · 2025-07-31
>
> **Q1:Limited Comparison with Prior Benchmarks: While the proposed benchmark is comprehensive, comparisons with other visual reasoning or security-related datasets are lacking, which limits understanding of relative difficulty and scope.**
>
> **A1**: Thanks! To better show the advantages of our proposed benchmarks, we compare some common visual reasoning benchmarks and security-related (intrusion detection) datasets from multiple different dimensions, i.e., task type and scene diversity, as shown in the table below. We can find that, compared to other benchmarks of visual reasoning or security-related tasks, our benchmark shows more comprehensive advantages, which meet the requirements of the proposed MLLM-ISU task. We are the first benchmark for intrusion scene understanding tasks.
>
> |Visual Reasonging Benchmark | Application Scene | Scene Diversity | Abnormal semantic information | Fine-grained understanding| Reasoning for safety | Task Type | Structure | Modal | Intrusion Labels| Intrusion Scene Understanding Capability |
> |:----|:----|:----|:----|:----|:----|:----|:----|:-----:|:-----:|:-----:|
> |FAVOR-Bench [30] | Video Motion Understanding | Low | No | Yes | No | Multi | Open | Multi-Modal  | No | Low |
> |MMAD [18] | Industrial anomaly detection| Low | Yes | No | No | Multi | Open | Multi-Modal  | No | Low |
> |MMID-bench [11] | Intrusion detection| High | Yes | Yes | Yes | Single | Close | Single-Modal  | Yes | Low |
> |MF-ID [10] | Intrusion detection| Low | Yes | Yes | Yes | Single | Close | Single-Modal | Yes | Low |
> |Ours | Industrial anomaly detection| High | Yes | Yes | Yes | Multi | Open | Multi-Modal  | Yes | High |
>
> **Q2:Data Bias Risk: Since the entire benchmark is based on Cityscapes, the generalizability to non-urban scenes or real-world surveillance data might be limited.**
>
> **A2**: Thanks! To enhance the diversity of intrusion scene types and improve the generalizability in real-world environments, we create a new benchmark dataset for the MMLM-ISU task, namely MMLM-ISU-BDD. The MMLM-ISU-BDD is built based on the BDD-100K datasets. The detailed method can refer to Fig. 3 in our paper. Our new MMLM-ISU-BDD datasets contain rich intrusion scene types, e.g., various weather conditions, geographic environments, periods of time, and transportation environments. We clean the original dataset based on the proposed intrusion detection task features. Finally, our datasets contain **8892** training Pairs and **2694** VQA evaluation Pairs. We evaluate the datasets on multiple Proprietary MLLMs, Open-source MLLMs, and Human performance, as shown in the table below. We can find that Current MLLMs are not capable of intrusion scene understanding, especially in harsh and volatile scenarios. Besides, we can also find that even the best models still have a wide gap with human performance. These phenomena suggest that there is still some way to go for the task of intrusion scene understanding. We will modify it in the final revision.
>
> |Model | Source | Release | IBJ | PIC | ISA | IOL | ICI | ISD | Average |
> |:----|:----|:----|:----|:----|:----|:----|:----|:----|:-----:|
> | Human | - | - | 90.00 | 96.67 | 38.89 | 85.56 | 96.67 | 35.61 | 73.90 |
> |GPT-4o [17] | OpenAI | 2024-08 | 19.38 |84.41 | 27.62| 62.14 | 89.98 | 6.40 | 48.32 |
> | LLaVa1.5-7B-hf [20] | UW-M&Micro | 2023-10 | 92.65 | 21.60 | 21.38 | 24.94 | 35.41 | 12.03 | 34.67 |
> | LLaVa1.5-13B-hf [20] | UW-M&Micro | 2023-10| 71.71 | 79.29 |21.60	|27.17|	32.52|	12.32|	40.77|
> | Qwen2-VL-7B-Instruct [32] | Alibaba | 2024-06 | 25.84 | 86.19 | 36.53 | 50.56 | 59.91 | 6.07 | 44.18 |
> | MiniCPM-V2.6 [35] | OpenBMB | 2024-08 | 5.79 | 83.96 | 30.29 | 33.85 | 77.95 | 12.44 | 40.71|
> | InternVL2.5-1B [4] |OpenGVLab | 2024-12 | 24.05 | 79.73 | 25.61 | 27.84 | 35.86 | 12.53| 34.27 |
> | InternVL2.5-2B [4] | OpenGVLab | 2024-12 | 5.35 | 83.96 | 30.73 | 45.66 | 43.88 | 11.69 | 36.88 |
> | InternVL2.5-8B [4] | OpenGVLab | 2024-12 | 24.05 | 77.73 | 29.84 | 61.92 | 66.37 | 13.80 | 45.62|
> |DeepSeek-VL2-tiny [33]|DeepSeek | 2024-12 | 73.05 | 86.64 | 30.73 | 51.22 | 76.61 | 18.25 | **56.08**|
> |DeepSeek-VL2-small [33]|DeepSeek|	2024-12	|32.07	|85.30	|27.39	|64.59	|79.51|	9.20|49.68|
> |Qwen2.5-VL-3B-Instruct [3]|Alibaba | 2025-01 | 35.41 | 77.73 | 29.18 | 89.76 | 80.62  | 5.79	| 53.08|
> |Qwen2.5-VL-7B-Instruct [3]|Alibaba|	2025-01|	18.49|	82.85|	26.50|	83.52|	86.41|	4.47	| 50.37|
> |Gemma3-4B-it [27] |Google|	2025-03|	49.00|	84.19|	22.49|	48.11|	32.07|	5.73|	40.27|
> |Gemma3-12B-it [27]|Google|	2025-03|	49.00|	43.65|	24.28|	71.94|	83.07|	5.60|	46.26|
> |Kimi-VL-A3B-Instruct [29]|Moonshot AI	| 2025-04|	16.93|	86.86|	26.50|	36.08|	88.42|	17.21|	45.33|
> |Kimi-VL-A3B-Thinking [29]|Moonshot AI |2025-04|	44.10|	20.27|	20.94|	24.50|	20.71|	5.08	|22.60|
> |InternVL3-2B [4]|OpenGVLab|	2025-04|	18.04|	75.72|	33.18|	42.32|	77.28|	6.57	|42.19|
> |InternVL3-8B [4]|OpenGVLab|	2025-04|	8.91|87.53|	25.17|	37.86|	42.76|	6.13	|34.73|
>
>
> **Q3:Model Scale: Some observations (e.g., smaller models outperforming larger ones) are reported but not deeply analyzed, and no concrete recommendation is given.**
>
> **A3**: Thanks! We agree that our previous version lacked an in-depth analysis of the observed phenomenon where smaller models (e.g., Qwen-VL-3B) sometimes outperform larger models (e.g., Qwen-VL-7B) in vision intrusion detection tasks. Upon further analysis, we find this can be attributed to two main factors:  (1) The first reason is that for the binary classification and region-level intrusion identification tasks, the model relies more on basic spatial perception than on complex reasoning, which leads to smaller models that may generalize better in such cases. (2) Besides, Larger models are more sensitive to prompt design. Our current prompt may favor concise, direct instructions, which smaller models handle better. We will clarify this analysis in section 5 of the revised manuscript.
>
> **Q4: Insufficient analysis of failure cases: While Section 5 illustrates that models may exhibit "overthinking" during multi-turn reasoning, it lacks an in-depth analysis of the failure causes. For instance, is the failure due to inadequate visual grounding during I-VIP, reasoning drift in I-COT, or hallucinated answers in I-VQA? A more systematic diagnosis would help clarify the limitations of each stage.**
>
> **A4**: Thanks! We further analyzed the types of model failures during the three-phase reasoning process and found that the main reasons can be summarized as: The main failure causes include inaccurate visual grounding in the I-VIP phase (e.g., small or overlapping objects), reasoning drift during complex inference in the I-COT phase, and hallucinated answers in the I-VQA phase due to ambiguous context or insufficient visual evidence. We will modify it in the revised manuscript.
>
> **Q5: Performance evaluation metrics: The paper lacks detailed explanation for the chosen evaluation metrics...., such as task-specific accuracy, human preference scores, or semantics-aware metrics?**
>
> **A5**: Thanks! (1) In our proposed task, low-level understanding tasks and high-level understanding tasks are designed as multiple-choice. Therefore, referring to previous work [18], we also set the metric to accuracy. This is because for multiple-choice questions, the answer is unique. Therefore, our accuracy rate is the optimal metric. Secondly, for open questions, we choose BLEU-4. We employ BLEU-4 as the primary metric for evaluating model-generated textual descriptions. In our benchmark, BLEU-4 helps to assess how well the model describes the visual content relevant to the question, especially in factual, descriptive, or object-level questions. We also give some other metrics.
>
> |Model |  | MLLM-ISU-CS | || MLLM-ISU-BDD |  |
> |:----:|:----:|:----:|:----:|:----:|:----:|:----:|
> |   | Rouge-1 | Rouge-2 | Rouge-L | Rouge-1 | Rouge-2 | Rouge-L |
> | GPT4o [17]  | 15.75 | 1.80 | 7.50 | 14.78 | 1.62 | 7.03 |
> | LLaVa1.5-13B-hf [6]  | 19.95 | 3.33 | 12.29 | 20.64 | 3.39 | 12.32 |
> | MiniCPM-V2.6 [11]  | 19.14 | 2.21 | 11.42 | 19.86 | 2.48 | 12.07 |
> | InternVL2.5-2B [3]  | 18.73 | 2.23 | 11.54 | 18.63 | 2.30 | 10.84 |
> | InternVL2.5-8B [3]  | 21.49 | 2.34 | 13.40 | 20.10 | 1.96 | 12.07 |
> | DeepSeek-VL2-tiny [10]  | 22.66 | 4.31 | 16.49 | 23.02 | 4.43 | 16.08 |
> | DeepSeek-VL2-small [10] | 20.41 | 2.32 | 8.60 | 19.09 | 1.65 | 8.98 |
> | Qwen2.5-VL-3B-Instruct [2]  | 16.44 | 1.72 | 6.94 | 14.61 | 1.53 | 6.19 |
> | Qwen2.5-VL-7B-Instruct [2]  | 14.74 | 1.64 | 5.07 | 13.75 | 1.57 | 4.86 |
> | Gemma3-4B-it [7]   | 14.56 | 1.92 | 5.31 | 13.82 | 1.74 | 5.07|
> | Gemma3-12B-it [7]  | 13.77 | 1.45 | 5.40 | 13.73 | 1.53 | 5.23 |
> | Kimi-VL-A3B-Instruct [9]  | 21.50 | 2.90 | 11.30 | 24.04 | 3.16 | 15.07 |
> | Kimi-VL-A3B-Thinking [9]  | 13.98 | 1.61 | 5.41 | 13.43 | 1.72 | 5.46 |
> | InternVL3-2B [3]  | 15.75 | 1.44 | 6.67 | 15.42 | 1.49 | 6.63 |
> | InternVL3-8B [3]  | 16.53 | 1.64 | 6.67 | 15.42| 1.65 | 6.21 |
>
> (2) Besides, we follow your suggestion and give some other evaluation results.
>
> **Human preference scores.** We give the human preference scores in MLLM-ISU tasks, as shown in the table below. We can find that even the best models (Qwen2.5-VL-3B-Instruct [3] and DeepSeek-VL2-tiny [33]) still have a wide gap with human performance in two different MLLM-ISU task benchmark datasets. This indicates a need to develop more effective methods to enhance the model's intrusion scene understanding capability. We will modify human performance and analyses in the final revision.
>
> |Model | Benchmark dataset | IBJ | PIC | ISA | IOL | ICI | ISD | Average |
> |:----|:----|:----|:----|:----|:----|:----|:----|:-----:|
> | Human | MLLM-ISU-CS | 98.00 | 72.00 | 44.00 | 77.00 | 98.00 | 39.36 | 71.39 |
> | Qwen2.5-VL-3B-Instruct [3] | MLLM-ISU-CS | 43.00 |  61.20 | 30.20 |  79.80 | 84.20 | 6.36 | 50.79 |
> | Human | MLLM-ISU-BDD | 90.00 | 96.67 | 38.89 | 85.56 | 96.67 | 35.61 | 73.90 |
> | DeepSeek-VL2-tiny [33] | MLLM-ISU-BDD | 73.05 | 86.64 | 30.73 | 51.22 | 76.61 | 18.25 | 56.08|

---

> ### Author Response · Authors · 2025-08-06
>
> **Dear Reviewer D4oT,**
>
> Thank you very much for the time and effort you have invested in reviewing our paper. We sincerely appreciate your thoughtful and constructive feedback, as well as your valuable contribution to the review process.
>
> We are especially grateful for your positive comments regarding our works and contributions: **(1)** Novel and Comprehensive Benchmark, **(2)** Systematic Training Pipeline, **(3)** Detailed Experimental Analysis, and **(4)** Novel Task Setting. The proposed benchmark and training pipeline significantly improve MLLM performance and offer valuable insights into model capability, generalization, and training dynamics.
>
> As the discussion phase has now passed its midpoint, we hope that our responses have sufficiently addressed your questions and concerns. Should there be any remaining points that require further clarification, please do not hesitate to let us know — we would be more than happy to provide additional explanation.
>
> With our warmest regards and sincere appreciation,
>
> **The Authors**

---

> ### Comment · Reviewer_D4oT · 2025-08-08
>
> I appreciate the author's detailed reply. The comprehensive explanation helps clarify the paper's claims, emphasizes its advantages over previous methods, and highlights the generalizability of the proposed approach in real-world environments. This addresses my question, and as a result, I will keep my original rating.

---

### Official Review · Reviewer_vTrG · 2025-07-02

**Rating:** 5
**Confidence:** 3

**Summary:**

This paper is motivated by the real-world complexity of interactive scenarios where multimodal reasoning, dialogue comprehension, and step-by-step task progression are essential. It presents the first benchmark for evaluating Interactive Scenario Understanding (ISU) using Multimodal Large Language Models (MLLMs), thereby filling a critical gap in current multimodal research.

**Dataset Code Accessibility:**

Yes

**Dataset Code Comments:**

The submission provides both the benchmark dataset and code used for experiments. All datasets, codes, and baselines will be publicly available. This makes it easy for researchers to replicate the reported results and build upon the work.

**Ethical Considerations:**

No, there are no or only very minor ethics concerns

**Final Justification:**

The authors have provided additional experiments to address my concerns.

**Limitations Weaknesses:**

1)The evaluation protocol is fully automated. Complementary human evaluations could help assess the nuances of interactive understanding more robustly.

2)Since this involves MLLMs over visual sequences and dialogue, details on processing costs and latency would help assess practical feasibility.

3)Many evaluated MLLMs are used in their default form. It would be informative to see performance with light task-specific adaptation to understand upper-bound performance.

**Strengths Contributions:**

1)The paper introduces a new and practically significant evaluation challenge — Interactive Scenario Understanding (ISU) — which requires MLLMs to combine multimodal reasoning with sequential interaction understanding.

2)A full benchmark is presented, including tailored datasets, well-defined evaluation metrics, and baseline models.

3)The paper proposes an automatic visual question-answer generation pipeline, enabling large-scale construction of interactive multimodal data. This significantly reduces the annotation burden and ensures scalability, especially since existing datasets are not readily compatible with the ISU setup.

---

> ### Author Rebuttal · Authors · 2025-07-31
>
> **Q1: The evaluation protocol is fully automated. Complementary human evaluations could help assess the nuances of interactive understanding more robustly.**
>
> **A1**: Good suggestions! To better evaluate the performance, we give comparisons to human performance in our paper. We invite someone, including PhDs, research assistants, and master's students, to test it, as shown in the table below. Note that MLLM-ISU-CS and MLLM-ISU-BDD denote the benchmark datasets that are built based on Cityscape and BDD-100K datasets for our MLLM-ISU task, respectively. We can find that even the best models (Qwen2.5-VL-3B-Instruct [3] and DeepSeek-VL2-tiny [33]) still have a wide gap with human performance in two different MLLM-ISU task benchmark datasets. This indicates a need to develop more effective methods to enhance the model's intrusion scene understanding capability. We will modify human performance and analyses in the final revision.
>
> |Model | Benchmark dataset | IBJ | PIC | ISA | IOL | ICI | ISD | Average |
> |:----|:----|:----|:----|:----|:----|:----|:----|:-----:|
> | Human | MLLM-ISU-CS | 98.00 | 72.00 | 44.00 | 77.00 | 98.00 | 39.36 | 71.39 |
> | Qwen2.5-VL-3B-Instruct [3] | MLLM-ISU-CS | 43.00 |  61.20 | 30.20 |  79.80 | 84.20 | 6.36 | 50.79 |
> | Human | MLLM-ISU-BDD | 90.00 | 96.67 | 38.89 | 85.56 | 96.67 | 35.61 | 73.90 |
> | DeepSeek-VL2-tiny [33] | MLLM-ISU-BDD | 73.05 | 86.64 | 30.73 | 51.22 | 76.61 | 18.25 | 56.08|
>
>
> **Q2: Since this involves MLLMs over visual sequences and dialogue, details on processing costs and latency would help assess practical feasibility.**
>
> **A2**: Thanks! We test the runtime, latency, or efficiency on two different models without any inference acceleration strategy, i.e., 3B and 7B. The input size of the image is **1024x2048**. As shown in the table below. We can find that our model has a low latency of 0.668 and a high predicted sample rate of 1.499. In the future, we will explore more efficient methods, i.e., Quantization, KV Cache, and FlashAttention, to further improve the performance for safety-critical applications.
>
>
> |Model | Task | Runtime in val (500 images) | Latency | Memory | Predicted_samples/s |
> |:----:|:----:|:----:|:----:|:----:|:----:|
> | 3B | IBJ | 5m34s | 0.668 | 9565MiB | 1.499 |
> | 3B | PIC | 5m46s | 0.692 | 9565MiB | 1.446 |
> | 7B | IBJ | 5m44s |0.688 | 18441MiB | 1.453 |
> | 7B | PIC | 5m47s |0.694 | 18441MiB | 1.443 |
>
>
> **Q3: Many evaluated MLLMs are used in their default form. It would be informative to see performance with light task-specific adaptation to understand upper-bound performance.**
>
> **A3**: Thanks! We explore a light task-specific strategy, e.g., prompt optimization, to understand upper-bound performance. We add a definitional prompt to facilitate better model understanding of the vision-based intrusion detection task, i.e., *Vision-based intrusion detection refers to judging whether a possible object exists in road.* We conduct experiments on two different MLLM-ISU datasets, MLLM-ISU-CS and MLLM-ISU-BDD.  Note that MLLM-ISU-CS and MLLM-ISU-BDD denote the benchmark datasets that are built based on Cityscape and BDD-100K datasets for our MLLM-ISU task, respectively. Besides, we test the light task-specific strategy on three MLLMs, i.e., Gemma3-4B [27], Kimi-VL-A3B-Instruct [29], InternVL3-2B [4], and report the average performance of six sub-tasks, as shown in the table below. We can find that our strategy can improve the upper-bound performance, achieving average performance gains of **1.27** to **4.07**, which proves the validity of our light task-specific strategy.
>
> |Model | Method | Average of six sub-task |Average of six sub-task|
> |:----:|:----:|:----:|:----:|
> | - |- | MLLM-ISU-CS | MLLM-ISU-BDD |
> | Gemma3-4B-it [27] | Zero-shot | 42.16 | 40.27 |
> |  | w/ task-specific prompting | 45.60 |43.14 |
> |  | Performance Gain | +3.44 | +2.87 |
> | Kimi-VL-A3B-Instruct [29] | Zero-shot | 42.14 | 45.33 |
> |  | w/ task-specific prompting | 44.38 |47.90 |
> |  | Performance Gain | +2.24 | +2.57 |
> | InternVL3-2B [4] | Zero-shot | 42.82 | 42.19 |
> |  | w/ task-specific prompting | 44.09 | 46.26 |
> |  | Performance Gain | +1.27 | +4.07 |

---

> > ### Comment · Reviewer_vTrG · 2025-08-06
> >
> > Thank you for the authors' detailed rebuttal and the additional experiments. They have adequately addressed my concerns, and I will increase my score to accept.

---

> ### Author Response · Authors · 2025-08-06
>
> **Dear Reviewer vTrG,**
>
> Thank you very much for the time and effort you have invested in reviewing our paper. We sincerely appreciate your thoughtful and constructive feedback, as well as your valuable contribution to the review process.
>
> We are especially grateful for your positive comments regarding our works and contributions: **(1)** introducing a new and practically significant evaluation challenge for the first time — Interactive Scenario Understanding (ISU), **(2)** presenting a full benchmark, including tailored datasets, well-defined evaluation metrics, and baseline models, and **(3)** proposing an automatic visual question-answer generation pipeline, enabling large-scale construction of interactive multimodal data. These contributions fill a critical gap in current multimodal research.
>
> As the discussion phase has now passed its midpoint, we hope that our responses have sufficiently addressed your questions and concerns. Should there be any remaining points that require further clarification, please do not hesitate to let us know — we would be more than happy to provide additional explanation.
>
> With our warmest regards and sincere appreciation,
>
> **The Authors**

---

### Official Review · Reviewer_RB3p · 2025-07-03

**Rating:** 5
**Confidence:** 2

**Summary:**

The paper introduces a novel benchmark and dataset—MLLM-ISU—targeting Intrusion Scene Understanding using Multimodal Large Language Models (MLLMs). The authors identify a research gap: while vision-based intrusion detection has been studied, comprehensive scene understanding in intrusion scenarios remains unexplored. To bridge this, they explore a novel task called Multimodal Large Language Models based Intrusion Scene Understanding and a comprehensive benchmark for the task.

**Dataset Code Accessibility:**

Partly

**Ethical Considerations:**

No, there are no or only very minor ethics concerns

**Final Justification:**

These revisions have substantially elevated the rigor, completeness, and impact of the paper. As a result, I am revising my rating to 5.

**Limitations Weaknesses:**

1.The benchmark is constructed mainly from the Cityscapes dataset, which may limit diversity in intrusion scene types (e.g., weather, nighttime, non-urban contexts), potentially affecting generalizability to more complex or real-world surveillance environments.

2. While quantitative results are thorough, the paper lacks qualitative error analysis or comparisons to human performance, which could provide better insight into the specific failure modes and what remains hard for MLLMs.

3. The framework is primarily focused on comprehension accuracy. However, no discussion is provided on runtime, latency, or efficiency, which are critical in intrusion detection systems deployed in safety-critical applications like autonomous driving or security monitoring.

**Strengths Contributions:**

1.This work introduces MLLM-ISU, the first benchmark tailored to evaluating Multimodal Large Language Models (MLLMs) on intrusion scene understanding tasks. It fills a major gap in the vision-language community by moving beyond object detection or semantic segmentation to focus on deeper comprehension of intrusion scenarios, including summarization, reasoning, and open-ended descriptions

2.The authors construct a large-scale dataset with 11,925 Visual Question Answering (VQA) pairs and define six distinct subtasks (e.g., Intrusion Behavior Judgment, Intrusion Scene Descriptions), covering low-level, high-level, and open-level understanding. The automatic question-answer generation pipeline combined with human verification ensures both scalability and quality.

3.The paper proposes a novel and interpretable three-stage fine-tuning strategy: Intrusion-aware Visual Instruction Pre-training, Chain-of-Thought Tuning, and Intrusion-centric VQA Tuning. Experiments show consistent improvements across multiple MLLMs, demonstrating the framework's effectiveness in boosting both reasoning and fine-grained perception abilities.

---

> ### Author Rebuttal · Authors · 2025-07-31
>
> **Q1: The benchmark is constructed mainly from the Cityscapes dataset, which may limit diversity in intrusion scene types (e.g., weather, nighttime, non-urban contexts), potentially affecting generalizability to more complex or real-world surveillance environments.**
>
> **A1:** Good suggestions! To enhance the diversity of intrusion scene types and improve the generalizability in real-world environments, we create a new benchmark dataset for the MMLM-ISU task, namely MMLM-ISU-BDD. The MMLM-ISU-BDD is built based on the BDD-100K datasets. The detailed method can refer to Fig. 3 in our paper. Our new MMLM-ISU-BDD datasets contain rich intrusion scene types, e.g., multiple different weather (Clear, Cloudy, Rainy, Foggy, Night), different geographic environment (City, Highway, Suburban/Rural), different period of time (Daytime, Dusk, Night), and Different transportation environments (Heavy Traffic, Empty Road). We clean the original dataset based on the proposed intrusion detection task features. Finally, our datasets contain **8892** training Pairs and **2694** VQA evaluation Pairs. We evaluate the datasets on multiple Proprietary MLLMs, Open-source MLLMs, and Human performance, as shown in the table below. We can find that Current MLLMs are not capable of intrusion scene understanding, especially in harsh and volatile scenarios. Besides, we can also find that even the best models still have a wide gap with human performance. These phenomena suggest that there is still some way to go for the task of intrusion scene understanding. Our extended benchmark explicitly includes nighttime, adverse weather, and non-urban roads, enabling more comprehensive evaluation of the intrusion scene understanding task in the real-world. We will modify it in the final revision.
>
> |Model | Source | Release | IBJ | PIC | ISA | IOL | ICI | ISD | Average |
> |:----|:----|:----|:----|:----|:----|:----|:----|:----|:-----:|
> | Human | - | - | 90.00 | 96.67 | 38.89 | 85.56 | 96.67 | 35.61 | 73.90 |
> |GPT-4o [17] | OpenAI | 2024-08 | 19.38 |84.41 | 27.62| 62.14 | 89.98 | 6.40 | 48.32 |
> | LLaVa1.5-7B-hf [20] | UW-M&Micro | 2023-10 | 92.65 | 21.60 | 21.38 | 24.94 | 35.41 | 12.03 | 34.67 |
> | LLaVa1.5-13B-hf [20] | UW-M&Micro | 2023-10| 71.71 | 79.29 |21.60	|27.17|	32.52|	12.32|	40.77|
> | Qwen2-VL-7B-Instruct [32] | Alibaba | 2024-06 | 25.84 | 86.19 | 36.53 | 50.56 | 59.91 | 6.07 | 44.18 |
> | MiniCPM-V2.6 [35] | OpenBMB | 2024-08 | 5.79 | 83.96 | 30.29 | 33.85 | 77.95 | 12.44 | 40.71|
> | InternVL2.5-1B [4] |OpenGVLab | 2024-12 | 24.05 | 79.73 | 25.61 | 27.84 | 35.86 | 12.53| 34.27 |
> | InternVL2.5-2B [4] | OpenGVLab | 2024-12 | 5.35 | 83.96 | 30.73 | 45.66 | 43.88 | 11.69 | 36.88 |
> | InternVL2.5-8B [4] | OpenGVLab | 2024-12 | 24.05 | 77.73 | 29.84 | 61.92 | 66.37 | 13.80 | 45.62|
> |DeepSeek-VL2-tiny [33]|DeepSeek | 2024-12 | 73.05 | 86.64 | 30.73 | 51.22 | 76.61 | 18.25 | **56.08**|
> |DeepSeek-VL2-small [33]|DeepSeek|	2024-12	|32.07	|85.30	|27.39	|64.59	|79.51|	9.20|49.68|
> |Qwen2.5-VL-3B-Instruct [3]|Alibaba | 2025-01 | 35.41 | 77.73 | 29.18 | 89.76 | 80.62  | 5.79	| 53.08|
> |Qwen2.5-VL-7B-Instruct [3]|Alibaba|	2025-01|	18.49|	82.85|	26.50|	83.52|	86.41|	4.47	| 50.37|
> |Gemma3-4B-it [27] |Google|	2025-03|	49.00|	84.19|	22.49|	48.11|	32.07|	5.73|	40.27|
> |Gemma3-12B-it [27]|Google|	2025-03|	49.00|	43.65|	24.28|	71.94|	83.07|	5.60|	46.26|
> |Kimi-VL-A3B-Instruct [29]|Moonshot AI	| 2025-04|	16.93|	86.86|	26.50|	36.08|	88.42|	17.21|	45.33|
> |Kimi-VL-A3B-Thinking [29]|Moonshot AI |2025-04|	44.10|	20.27|	20.94|	24.50|	20.71|	5.08	|22.60|
> |InternVL3-2B [4]|OpenGVLab|	2025-04|	18.04|	75.72|	33.18|	42.32|	77.28|	6.57	|42.19|
> |InternVL3-8B [4]|OpenGVLab|	2025-04|	8.91|87.53|	25.17|	37.86|	42.76|	6.13	|34.73|
>
> **Q2:While quantitative results are thorough, the paper lacks qualitative error analysis or comparisons to human performance, which could provide better insight into the specific failure modes and what remains hard for MLLMs.**
>
> A2: Good suggestions! To better evaluate the performance, we give comparisons to human performance in our paper. We invite someone, including PhDs, research assistants, and master's students, to test it, as shown in the table below. Note that MLLM-ISU-CS and MLLM-ISU-BDD denote the benchmark datasets that are built based on Cityscape and BDD-100K datasets for our MLLM-ISU task, respectively. We can find that even the best models (Qwen2.5-VL-3B-Instruct [3] and DeepSeek-VL2-tiny [33]) still have a wide gap with human performance in two different MLLM-ISU task benchmark datasets. This indicates a need to develop more effective methods to enhance the model's intrusion scene understanding capability. We will modify human performance and analyses in the final revision.
>
> |Model | Benchmark dataset | IBJ | PIC | ISA | IOL | ICI | ISD | Average |
> |:----|:----|:----|:----|:----|:----|:----|:----|:-----:|
> | Human | MLLM-ISU-CS | 98.00 | 72.00 | 44.00 | 77.00 | 98.00 | 39.36 | 71.39 |
> | Qwen2.5-VL-3B-Instruct [3] | MLLM-ISU-CS | 43.00 |  61.20 | 30.20 |  79.80 | 84.20 | 6.36 | 50.79 |
> | Human | MLLM-ISU-BDD | 90.00 | 96.67 | 38.89 | 85.56 | 96.67 | 35.61 | 73.90 |
> | DeepSeek-VL2-tiny [33] | MLLM-ISU-BDD | 73.05 | 86.64 | 30.73 | 51.22 | 76.61 | 18.25 | 56.08|
>
> **Q3: The framework is primarily focused on comprehension accuracy. However, no discussion is provided on runtime, latency, or efficiency, which are critical in intrusion detection systems deployed in safety-critical applications like autonomous driving or security monitoring.**
>
> A3: Thanks! We test the runtime, latency, or efficiency on two different models without any inference acceleration strategy, i.e., 3B and 7B. The input size of the image is **1024x2048**. As shown in the table below. We can find that our model has a low latency of 0.668 and a high predicted sample rate of 1.499. In the future, we will explore more efficient methods, i.e., quantization, KV Cache, and flashAttention, to further improve the performance for safety-critical applications.
>
>
> |Model | Task | Runtime in val (500 images) | Latency | Memory | Predicted_samples/s |
> |:----:|:----:|:----:|:----:|:----:|:----:|
> | 3B | IBJ | 5m34s | 0.668 | 9565MiB | 1.499 |
> | 3B | PIC | 5m46s | 0.692 | 9565MiB | 1.446 |
> | 7B | IBJ | 5m44s |0.688 | 18441MiB | 1.453 |
> | 7B | PIC | 5m47s |0.694 | 18441MiB | 1.443 |

---

> > ### Comment · Reviewer_RB3p · 2025-08-05
> >
> > Thank you for your comprehensive rebuttal and the significant improvements made to address my earlier concerns. I am particularly impressed by the steps taken to strengthen the work: expanding the benchmark with MLLM-ISU-BDD introduces critical diversity in scenarios—including varied weather, times of day, and non-urban contexts—directly enhancing the generalizability of the dataset, which was a key gap I highlighted. The addition of human performance comparisons provides valuable insight into the current limitations of MLLMs, clearly illustrating the gap between model performance and human understanding, especially in complex subtasks like ISA and ISD, and this context enriches the analysis of the task’s challenges. Furthermore, incorporating runtime, latency, and memory metrics addresses a crucial practical consideration for safety-critical applications, with the preliminary results (e.g., 0.668s latency for the 3B model) offering a solid baseline, and your plan to explore acceleration techniques further underscores the work’s real-world relevance. These revisions have substantially elevated the rigor, completeness, and impact of the paper. As a result, I am revising my rating to 5.

---

### Note · Authors · 2025-08-12

We sincerely thank all reviewers, AC, SAC, and PC for their valuable efforts and constructive feedback. We appreciate the reviewers' recognition of our work's contributions, including **the first and novel benchmark for intrusion scene understanding tasks** (Reviewers RB3p, vTrG, D4oT, ePVt), **the efficient automatic visual question-answer generation pipeline** (Reviewers RB3p, vTrG, ePVt), **the effective and interpretable three-stage fine-tuning strategy** (Reviewers RB3p, D4oT, ePVt),  **the detailed and comprehensive experimtns and evaluation** (Reviewers RB3p, D4oT, ePVt), **filling a critical gap in current multimodal research** (Reviewer vTrG), **valuable and original contribution** (Reviewer D4oT), **the novel and interesting research**  (Reviewer ePVt).

---

During the rebuttal and discussion phases, we provided detailed point-by-point responses addressing most of the reviewers' concerns, mainly covering:

(1) *Constructing new datasets* to enhance the diversity of intrusion scene types and improve the generalizability in complex, real-world environments.

(2) *Incorporating comparisons with human performance* to more comprehensively evaluate the subtleties of interactive scene understanding.

(3) *Conducting thorough runtime, latency, and efficiency* evaluations of our model for intrusion detection to demonstrate practical applicability.

(4) *Designing a lightweight, task-specific adaptation strategy* to explore the upper-bound potential of our approach.

(5) *Providing more in-depth analysis and comparative* to clarify further and support the contributions of our work.

**Following the reviewers' suggestions, we will carefully and thoroughly address each comment in the revised version.**

---

We believe MLLM-ISU makes critical contributions to the community: it provides a **first and comprehensive** benchmark for the intrusion scene understanding task. We introduce an effective automatic visual question-answer generation pipeline, overcoming the limitation that existing datasets are not directly applicable to the MLLM-ISU task. Besides, an effective post-training framework with sequential training strategies is designed to improve the performance of the MLLM-ISU task. Extensive experiments show that our benchmark and training pipeline significantly improve MLLM performance and offer valuable insights.

We again thank all reviewers, AC, SAC, and PC, for the time and effort devoted to reviewing our work and helping us improve this paper.

---

### Decision · Program_Chairs · 2025-09-18

**Decision:**

Accept (poster)

**Comment:**

This paper introduces MLLM-ISU, the first comprehensive benchmark for Multimodal Large Language Models in intrusion scene understanding. The dataset is generated using an automated visual question-answering strategy and features a multi-stage post-training framework.

The paper received 4 reviews, with two "Accept" and two "Borderline Accept".

Strengths: The work is the first benchmark for intrusion scene understanding tasks. The automatic question-answer generation pipeline combined with human verification ensures both scalability and quality. The proposed three-stage training framework effectively boosts model performance, providing a strong baseline.

Main Concerns: The dataset relies heavily on cityscapes, which may limit its generalization to different environments. The dataset lacks in-depth qualitative analysis, comparison with human performance, and discussion of practical application metrics such as latency.

The authors provided a convincing rebuttal that fully addressed the reviewers' concerns. They introduced a new dataset (MLLM-ISU-BDD) to enhance diversity, added a human performance comparison, and promised to include a discussion on efficiency. The AC therefore recommends acceptance.